# GeoMFormer: A General Architecture for Geometric Molecular Representation Learning

## Abstract

Molecular modeling, a central topic in quantum mechanics, aims to accurately calculate the properties and simulate the behaviors of molecular systems. The molecular model is governed by physical laws, which impose geometric constraints such as invariance and equivariance to coordinate rotation and translation. While numerous deep learning approaches have been developed to learn molecular representations under these constraints, most of them are built upon heuristic and costly modules. We argue that there is a strong need for a general and flexible framework for learning both invariant and equivariant features. In this work, we introduce a novel Transformer-based molecular model called GeoMFormer to achieve this goal. Using the standard Transformer modules, two separate streams are developed to maintain and learn invariant and equivariant representations. Carefully designed *cross-attention* modules bridge the two streams, allowing information fusion and enhancing geometric modeling in each stream. As a general and flexible architecture, we show that many previous architectures can be viewed as special instantiations of GeoMFormer. Extensive experiments are conducted to demonstrate the power of GeoMFormer. All empirical results show that GeoMFormer achieves strong performance on both invariant and equivariant tasks of different types and scales. Code and models will be made publicly available.

## 1 Introduction

Deep learning approaches have emerged as a powerful tool for a wide range of tasks, such as image classification and language understanding (He et al., 2016; Devlin et al., 2019; Brown et al., 2020). Recently, researchers have started investigating whether the power of neural networks could help solve problems in physics and chemistry, such as predicting the property of molecules with 3D coordinates and simulating how each atom moves in Euclidean space (Schütt et al., 2018; Gasteiger et al., 2020b; Satorras et al., 2021). These molecular modeling tasks require the learned model to satisfy general physical laws, such as the invariance and equivariance conditions: The model's prediction should react *physically* when the input coordinates change according to the transformation of the coordinate system, such as rotation and translation.

A variety of methods have been proposed to design neural architectures that intrinsically satisfy the invariance or equivariance conditions (Thomas et al., 2018; Schütt et al., 2021; Batzner et al., 2022). To satisfy the invariant condition, several approaches incorporate invariant features, such as the relative distance between each atom pair, into classic neural networks (Schütt et al., 2018; Shi et al., 2022). However, this may hinder the model from effectively extracting the molecular structural information. For example, computing dihedral angles from coordinates is straightforward but requires much more nonlinear operations using relative distances. To satisfy the equivariant condition, several works design neural networks with equivariant operation only, such as tensor product between irreducible representations (Thomas et al., 2018; Fuchs et al., 2020; Batzner et al., 2022) and vector operations (Satorras et al., 2021; Schütt et al., 2021; Thölke & De Fabritiis, 2022). However, the number of such operations is limited, and they are either costly to scale or lead to fairly complex network architecture designs to guarantee sufficient expressive power. More importantly, many real-world applications require a model that can effectively perform both invariant and equivariant prediction with strong performance at the same time. While some recent works study this direction (Schütt et al., 2021; Thölke & De Fabritiis, 2022), most proposed networks are designed heuristically and lack general design principles.

We argue that developing a general and flexible architecture to effectively learn both invariant and equivariant representations is essential. In this work, we introduce GeoMFormer to achieve this goal. GeoMFormer uses a standard Transformer-based architecture (Vaswani et al., 2017) but with two streams. An invariant stream learns invariant representations, and an equivariant stream learns

equivariant representations. Each stream is composed of invariant/equivariant self-attention and feed-forward layers. The key design in GeoMFormer is to use *cross-attention mechanisms* between the two streams, letting each stream incorporate the information from the other and enhance itself. In each layer of the invariant stream, we develop an *invariant-to-equivariant* cross-attention module, where the invariant representations are used to query key-value pairs in the equivariant stream. An *equivariant-to-invariant* cross-attention module is designed similarly in the equivariant stream. We show that the design of all self-attention and cross-attention modules is flexible and how to satisfy the invariant/equivariant conditions effectively.

Our proposed architecture has several advantages compared to previous works. GeoMFormer decomposes the invariant/equivariant representation learning through self-attention and cross-attention modules. By interacting the two streams using cross-attention modules, the invariant stream receives more structural signals (from the equivariant stream), and the equivariant stream obtains more non-linear transformation (from the invariant stream), which allows simultaneously and completely modeling interatomic interactions within/across feature spaces in a unified manner. Furthermore, we demonstrate that the proposed decomposition is general by showing that many existing methods can be regarded as special cases in our framework. For example, PaiNN(Schütt et al., 2021) and TorchMD-NET(Thölke & De Fabritiis, 2022) can be formulated as a special instantiation by following the design philosophy of GeoMFormer and using proper instantiations of key building components. From this perspective, we believe our architecture can offer many different options in different scenarios in real applications.

We evaluate our architecture on diverse datasets with both invariant and equivariant targets. On the Open Catalyst 2020 (OC20) dataset (Chanussot et al., 2021), which contains large atomic systems composed of an adsorbate and a catalyst, our architecture is able to predict the system's energy (invariant) and relaxed structure (equivariant) with high accuracy. Additionally, our architecture achieves state-of-the-art performance for predicting homo-lumo energy gap (invariant) of a molecule on PCQM4Mv2 (Hu et al., 2021) and Molecule3D (Xu et al., 2021) datasets, both of which consist of molecules collected from the chemical database (Maho, 2015; Nakata & Shimazaki, 2017). Moreover, we conduct an N-body simulation experiment where our architecture can precisely forecast the positions (equivariant) for a set of particles controlled by physical rules. All the empirical results highlight the generality and effectiveness of our approach.

## 2 RELATED WORKS

**Invariant Representation Learning.** In recent years, invariance has been recognized as one of the fundamental principles guiding the development of molecular models. To describe the properties of a molecular system, the model's prediction should remain unchanged if we conduct any rotation or translation actions on the coordinates of the whole system. Previous works usually rely on relative structural signals from the coordinates, which intrinsically preserve the invariance. In SchNet (Schütt et al., 2018), the interatomic distances are encoded via radial basis functions, which serve as the weights of the developed continuous-filter convolutional layers. PhysNet (Unke & Meuwly, 2019) similarly incorporated both atomic features and interatomic distances in its interaction blocks. Graphormer-3D (Shi et al., 2022) developed a Transformer-based model by encoding the relative distance as attention bias terms, which perform well on large-scale datasets (Chanussot et al., 2021).

Beyond the interatomic distance, other works further incorporate high-order invariant signals. Based on PhysNet, DimeNet (Gasteiger et al., 2020b) and DimeNet++ (Gasteiger et al., 2020a) additionally encode the bond angle information using Fourier-Bessel basis functions. Moreover, GemNet (Gasteiger et al., 2021) and GemNet-OC (Gasteiger et al., 2022) carefully studied the connections between spherical representations and directional information, which inspired to leverage the dihedral angles, i.e., angles between planes formed by bonds. SphereNet (Liu et al., 2022b) and ComENet (Wang et al., 2022) consider the torsional information to augment the molecular models. During the development in the literature, more complex features are incorporated due to the lossy structural information when purely learning invariant representations, while largely increasing the costs. Moreover, these invariant models are generally unable to directly perform equivariant prediction tasks.

**Equivariant Representation Learning.** Instead of building invariant blocks only, there are various works that aim to learn equivariant representations. In real-world applications, there are also many molecular tasks that require the model to perform equivariant predictions, e.g., predicting the force, position, velocity, and other tensorized properties in dynamic simulation tasks. If a rotation action is performed on each position, then these properties should also correspondingly rotate. One classical

approach (Thomas et al., 2018; Fuchs et al., 2020; Batzner et al., 2022; Musaelian et al., 2023) to encoding the equivariant constraints is using irreducible representations (irreps) via spherical harmonics (Goodman & Wallach, 2000). With equivariant convolutions based on tensor products between irreps, each block of the model preserves the equivariance. However, these models do not always significantly outperform invariant models on invariant tasks. Besides, their operations are in general costly (Schütt et al., 2021; Satorras et al., 2021; Frank et al., 2022), which largely hinders the model from deploying on large-scale molecular systems.

On the other hand, several recent works maintain both invariant and equivariant representations. To this end, EGNN (Satorras et al., 2021) proposed a simple framework. Its invariant representations encode type information and relative distance, and are further used in vector scaling functions to transform the equivariant representations. PaiNN (Schütt et al., 2021) extended the framework of EGNN to include the Hardamard product operation to transform the equivariant representations. Based on the operations of PaiNN, TorchMD-Net (Thölke & De Fabritiis, 2022) further proposed a modified version of the self-attention modules to update invariant representations and achieved better performance on invariant tasks. Allegro (Musaelian et al., 2023) instead uses tensor product operations to update equivariant features and interacts equivariant and invariant features by using weight-generation modules. In contrast, our GeoMFormer is developed based on a general design philosophy to learn both invariant and equivariant representations, which enables simultaneously and completely modeling interatomic interactions within/across feature spaces in a unified manner. as introduced in Section 4.1.

## 3 PRELIMINARY

### 3.1 NOTATIONS & GEOMETRIC CONSTRAINTS

We denote a molecular system as $\mathcal{M}$, which is made up of a collection of atoms held together by attractive forces. Let $\mathbf{X} \in \mathbb{R}^{n \times d}$ denote the atoms with features, where $n$ is the number of atoms, and $d$ is the feature dimension. Given atom $i$, we use $\mathbf{r}_i \in \mathbb{R}^3$ to denote its cartesian coordinate in the three-dimensional Euclidean space. We define $\mathcal{M} = (\mathbf{X}, R)$, where $R = \{\mathbf{r}_1, ..., \mathbf{r}_n\}$.

In nature, molecular systems are subject to physical laws that impose geometric constraints on their properties and behaviors. For instance, if the position of each atom in a molecular system is translated by a constant vector in Euclidean space, the total energy of the system remains unchanged. If a rotation is applied to each position, the direction of the force on each atom will also rotate. Mathematically, these geometric constraints are directly related to the concepts of invariance and equivariance in group theory (Cotton, 1991; Cornwell, 1997; Scott, 2012).

Formally, let $\phi : \mathcal{X} \to \mathcal{Y}$ denote a function mapping between vector spaces. Given a group $G$, let $\rho^{\mathcal{X}}$ and $\rho^{\mathcal{Y}}$ denote its group representations. A function $\phi : \mathcal{X} \to \mathcal{Y}$ is said to be equivariant/invariant if it satisfies the following conditions respectively:

$$
\begin{aligned}
\textit{Equivariance}: \quad & \rho^{\mathcal{Y}}(g)[\phi(x)] = \phi\left(\rho^{\mathcal{X}}(g)[x]\right), \text{ for all } g \in G, x \in \mathcal{X} \\
\textit{Invariance}: \quad & \phi(x) = \phi\left(\rho^{\mathcal{X}}(g)[x]\right), \text{ for all } g \in G, x \in \mathcal{X}
\end{aligned}
\tag{1}
$$

Intuitively, an equivariant function mapping transforms the output predictably in response to transformations on the input, whereas an invariant function mapping produces an output that remains unchanged by transformations applied to the input. For further details on the background of group theory, we refer readers to the appendix of (Thomas et al., 2018; Anderson et al., 2019; Fuchs et al., 2020).

Molecular systems are naturally located in the three-dimensional Euclidean space, and the group related to translations and rotations is known as $SE(3)$. For each element $g$ in the $SE(3)$ group, its representation on $\mathbb{R}^3$ can be parameterized by pairs of translation vectors $\mathbf{t} \in \mathbb{R}^3$ and orthogonal transformation matrices $\mathbf{R} \in \mathbb{R}^{3 \times 3}, \det(\mathbf{R}) = 1$, i.e., $g = (\mathbf{t}, \mathbf{R})$. Given a vector $x \in \mathbb{R}^3$, we have $\rho^{\mathbb{R}^3}(g)[x] := \mathbf{R}x + \mathbf{t}$. For molecular modeling, it is essential to learn molecular representations that encode the rotation equivariance and translation invariance constraints. Formally, let $V_{\mathcal{M}}$ denote the space of molecular systems, for each atom $i$, we define equivariant representation $\phi^E$ and invariant representation $\phi^I$ if $\forall\, g = (\mathbf{t}, \mathbf{R}) \in SE(3), \mathcal{M} = (\mathbf{X}, R) \in V_{\mathcal{M}}$, the following conditions are satisfied:

$$
\begin{aligned}
\phi^E &: V_{\mathcal{M}} \to \mathbb{R}^{3 \times d}, & \mathbf{R}\phi^E(\mathbf{X}, \{\mathbf{r}_1, ..., \mathbf{r}_n\}) &= \phi^E(\mathbf{X}, \{\mathbf{R}\mathbf{r}_1, ..., \mathbf{R}\mathbf{r}_n\}) \\
\phi^E &: V_{\mathcal{M}} \to \mathbb{R}^{3 \times d}, & \phi^E(\mathbf{X}, \{\mathbf{r}_1, ..., \mathbf{r}_n\}) &= \phi^E(\mathbf{X}, \{\mathbf{r}_1 + \mathbf{t}, ..., \mathbf{r}_n + \mathbf{t}\}) \\
\phi^I &: V_{\mathcal{M}} \to \mathbb{R}^{d}, & \phi^I(\mathbf{X}, \{\mathbf{r}_1, ..., \mathbf{r}_n\}) &= \phi^I(\mathbf{X}, \{\mathbf{R}\mathbf{r}_1 + \mathbf{t}, ..., \mathbf{R}\mathbf{r}_n + \mathbf{t}\})
\end{aligned}
\tag{2}
$$

## 3.2 ATTENTION MODULE

The attention module lies at the core of the Transformer architecture (Vaswani et al., 2017), and it is formulated as querying a dictionary with key-value pairs, e.g., $\text{Attention}(Q, K, V) = \text{softmax}(\frac{QK^T}{\sqrt{d}})V$, where $d$ is the hidden dimension, and $Q$ (Query), $K$ (Key), $V$ (Value) are specified as the hidden representations of the previous layer. The multi-head variant of the attention module is widely used, as it allows the model to jointly attend to information from different representation subspaces. It is defined as follows:

$$\text{Multi-head}(Q, K, V) = \text{Concat}(\text{head}_1, \cdots, \text{head}_H)W^O$$
$$\text{head}_k = \text{Attention}(QW_k^Q, KW_k^K, VW_k^V), \tag{3}$$

where $W_k^Q \in \mathbb{R}^{d \times d_H}, W_k^K \in \mathbb{R}^{d \times d_H}, W_k^V \in \mathbb{R}^{d \times d_H}$, and $W^O \in \mathbb{R}^{Hd_H \times d}$ are learnable matrices, $H$ is the number of heads. $d_H$ is the dimension of each attention head.

Serving as a generic building block, the attention module can be used in various ways. On the one hand, the self-attention module specifies Query, Key, and Value as the same hidden representation, thereby extracting contextual information for the input. It has been one of the key components in Transformer-based foundation models across various domains (Devlin et al., 2019; Brown et al., 2020; Dosovitskiy et al., 2021; Liu et al., 2021; Ying et al., 2021a; Jumper et al., 2021). On the other hand, the cross-attention module specifies the hidden representation from one space as Query, and the representation from the other space as Key-Value pairs, e.g. encoder-decoder attention for sequence-to-sequence learning. As the cross-attention module bridges two representation spaces, it has been also widely used beyond Transformer for information fusion and improving representations (Lee et al., 2018; Huang et al., 2019; Jaegle et al., 2021b;a).

## 4 GEOMFORMER

In this section, we introduce GeoMFormer, a novel Transformer-based molecular model for learning invariant and equivariant molecular representations. We begin by elaborating on the key designs of GeoMFormer, which form a general framework to guide the development of geometric molecular models (Section 4.1), Next we thoroughly discuss the implementation details of GeoMFormer (Section 4.2).

### 4.1 A GENERAL DESIGN PHILOSOPHY

As previously mentioned, several existing works learned invariant representations using invariant features, such as distance information, which may have difficulty in extracting other useful structural signals. Some other works developed equivariant models via equivariant operations, which are either heuristic or costly. Instead, we aim to develop a general design principle, which guides the development of a model instance that addresses the disadvantages aforementioned in both invariant and equivariant representation learning.

We call our model GeoMFormer, which is a two-stream Transformer model to encode invariant and equivariant information. Each stream is built up using stacked Transformer blocks, each of which consists of a self-attention module and a cross-attention module, followed by a feed-forward network. For each atom $k \in [n]$, we use $\mathbf{z}_k^I \in \mathbb{R}^d$ and $\mathbf{z}_k^E \in \mathbb{R}^{3 \times d}$ to denote its invariant and equivariant representations respectively. Let $\mathbf{Z}^I = [\mathbf{z}_1^{I^\top}; ...; \mathbf{z}_n^{I^\top}] \in \mathbb{R}^{n \times d}$ and $\mathbf{Z}^E = [\mathbf{z}_1^E; ...; \mathbf{z}_n^E] \in \mathbb{R}^{n \times 3 \times d}$, the invariant (colored in red) and equivariant (colored in blue) representations are updated in the following manner:

$$
\begin{aligned}
\textit{Invariant Stream} &\begin{cases}
\mathbf{Z}'^{I,l} &= \mathbf{Z}^{I,l} + \text{Inv-Self-Attn}(\mathbf{Q}^{I,l}, \mathbf{K}^{I,l}, \mathbf{V}^{I,l}) \\
\mathbf{Z}''^{I,l} &= \mathbf{Z}'^{I,l} + \text{Inv-Cross-Attn}(\mathbf{Q}^{I,l}, \mathbf{K}^{I\_E,l}, \mathbf{V}^{I\_E,l}) \\
\mathbf{Z}^{I,l+1} &= \mathbf{Z}''^{I,l} + \text{Inv-FFN}(\mathbf{Z}''^{I,l})
\end{cases} \\[2ex]
\textit{Equivariant Stream} &\begin{cases}
\mathbf{Z}'^{E,l} &= \mathbf{Z}^{E,l} + \text{Equ-Self-Attn}(\mathbf{Q}^{E,l}, \mathbf{K}^{E,l}, \mathbf{V}^{E,l}) \\
\mathbf{Z}''^{E,l} &= \mathbf{Z}'^{E,l} + \text{Equ-Cross-Attn}(\mathbf{Q}^{E,l}, \mathbf{K}^{E\_I,l}, \mathbf{V}^{E\_I,l}) \\
\mathbf{Z}^{E,l+1} &= \mathbf{Z}''^{E,l} + \text{Equ-FFN}(\mathbf{Z}''^{E,l})
\end{cases}
\end{aligned} \tag{4}
$$

where $l$ denotes the layer index. In this framework, the self-attention modules and feed-forward networks are used to iteratively update representations in each stream. The cross-attention modules use representations from one stream to query Key-Value pairs from the other stream. By using

this mechanism, a bidirectional bridge is established between invariant and equivariant streams. Besides the contextual information from the invariant stream itself, the invariant representations can freely attend to more geometrical signals from the equivariant stream. Similarly, the equivariant representations can benefit from using more non-linear transformations in the invariant representations. With the cross-attention modules, the expressiveness of both invariant and equivariant representation learning is largely improved, which allows simultaneously and completely modeling interatomic interactions within/across feature spaces in a unified manner. In this regard, as highlighted by different colors, the Query, Key, and Value in the self-attention modules (Inv-Self-Attn, Equ-Self-Attn) and the cross-attention modules (Inv-Cross-Attn, Equ-Cross-Attn) are differently specified, which should carefully encode the geometric constraints mentioned in Section 3.1, as introduced below.

**Desiderata for Invariant Self-Attention.** Given the invariant representation $\mathbf{Z}^I$, the Query, Key and Value in Inv-Self-Attn are calculated via a function mapping $\psi^I : \mathbb{R}^{n \times d} \to \mathbb{R}^{n \times d}$, i.e., $\mathbf{Q}^I = \psi_Q^I(\mathbf{Z}^I), \mathbf{K}^I = \psi_K^I(\mathbf{Z}^I), \mathbf{V}^I = \psi_V^I(\mathbf{Z}^I)$. Essentially, the attention module linearly transforms the Value $\mathbf{V}^I$, with the weights being calculated from the dot product between the Query and Key (i.e., attention scores). In this regard, if both $\mathbf{V}^I$ and the attention scores preserve the invariance, then the output satisfies the invariant constraint, i.e., $\psi^I$ is required to be invariant. Under this condition, it is easy to check the output representation of this module keeps the invariance, which is proved in the appendix.

**Desiderata for Equivariant Self-Attention.** Similarly, given the equivariant input $\mathbf{Z}^E$, the Query, Key and Value in Equ-Self-Attn are calculated via a function mapping $\psi^E : \mathbb{R}^{n \times 3 \times d} \to \mathbb{R}^{n \times 3 \times d}$, i.e., $\mathbf{Q}^E = \psi_Q^E(\mathbf{Z}^E), \mathbf{K}^E = \psi_K^E(\mathbf{Z}^E), \mathbf{V}^E = \psi_V^E(\mathbf{Z}^E)$. Similarly, $\psi^E$ is required to be equivariant. However, this still cannot guarantee the module to be equivariant if standard attention is used. We modified $\alpha_{ij} = \sum_{k=1}^d \mathbf{Q}^E_{[i,:,k]} \mathbf{K}^E_{[j,:,k]}{}^\top$, where $\mathbf{Q}^E_{[i,:,k]} \in \mathbb{R}^3$ denotes the $k$-th dimension of the atom $i$'s Query. It is straightforward to check the equivariance is preserved, which is proved in the appendix.

**Desiderata for Cross-attentions between the two Streams.** In each stream, the cross-attention module is used to leverage information from the other stream. We call the cross attention in the invariant stream *invariant-cross-equivariant* attention, and call the cross attention in the equivariant stream *equivariant-cross-invariant* attention, i.e., Inv-Cross-Attn and Equ-Cross-Attn. The difference between the two cross attention lies in how the query, key, value are specified:

$$
\begin{aligned}
\textit{Invariant-cross-Equivariant} \quad & \mathbf{Q}^{I\_E} = \psi_Q^I(\mathbf{Z}^I), \mathbf{K}^{I\_E} = \psi_K^{I\_E}(\mathbf{Z}^I, \mathbf{Z}^E), \mathbf{V}^{I\_E} = \psi_V^{I\_E}(\mathbf{Z}^I, \mathbf{Z}^E) \\
\textit{Equivariant-cross-Invariant} \quad & \mathbf{Q}^{E\_I} = \psi_Q^E(\mathbf{Z}^E), \mathbf{K}^{E\_I} = \psi_K^{E\_I}(\mathbf{Z}^E, \mathbf{Z}^I), \mathbf{V}^{E\_I} = \psi_V^{E\_I}(\mathbf{Z}^E, \mathbf{Z}^I)
\end{aligned}
\tag{5}
$$

First, for Query $\mathbf{Q}^{I\_E}$ and $\mathbf{Q}^{E\_I}$, the requirement to $\psi^I$ and $\psi^E$ remains the same as previously stated. Moreover, as distinguished by different colors, the Key-Value pairs and the Query are calculated in different ways, for which the requirement should be separately considered. Note that both $\mathbf{V}^{I\_E}$ and $\mathbf{V}^{E\_I}$ are still linearly transformed by the cross-attention modules. If $\mathbf{V}^{I\_E}$ preserves the invariance and $\mathbf{V}^{E\_I}$ preserves the equivariance, then the remaining condition is to keep the invariance of the attention score calculation. That is to say, for the Inv-Cross-Attn, both $\psi^I$ and $\psi^{I\_E}$ are required to be invariant. It is similar to the Equ-Cross-Attn that both $\psi^E$ and $\psi^{E\_I}$ are required to be equivariant. In this way, the outputs of both cross-attention modules are under the corresponding geometric constraints, which is proved in the appendix.

**Discussion.** The carefully designed blocks outlined above provide a general design philosophy for encoding the geometric constraints and bridging the invariant and equivariant molecular representations, which lie at the core of our framework. Note that the translation invariance can be easily preserved by encoding relative structure signals of the input. It is also worth pointing out that we do not restrict the specific instantiation of each component, and various design choices can be adopted as long as they meet the requirements mentioned above. Moreover, we prove that our framework can include many previous models as an instantiation, e.g., PaiNN (Schütt et al., 2021) and TorchMD-Net (Thölke & De Fabritiis, 2022), can be extended to encode additional geometric constraints (Cornwell, 1997), which are presented in the appendix. In this work, we present a simple yet effective model instance that implements this design philosophy, which we will thoroughly introduce in the next subsection.

## 4.2 IMPLEMENTATION DETAILS OF GEOMFORMER

Following the design guidance in Section 4.1, we propose **Geo**metric **M**olecular Transformer (GeoMFormer). The overall architecture of GeoMFormer is shown in Figure 1, which is composed of stacked GeoMFormer blocks (Eqn.(5)). We introduce the instantiations of the self-attention,

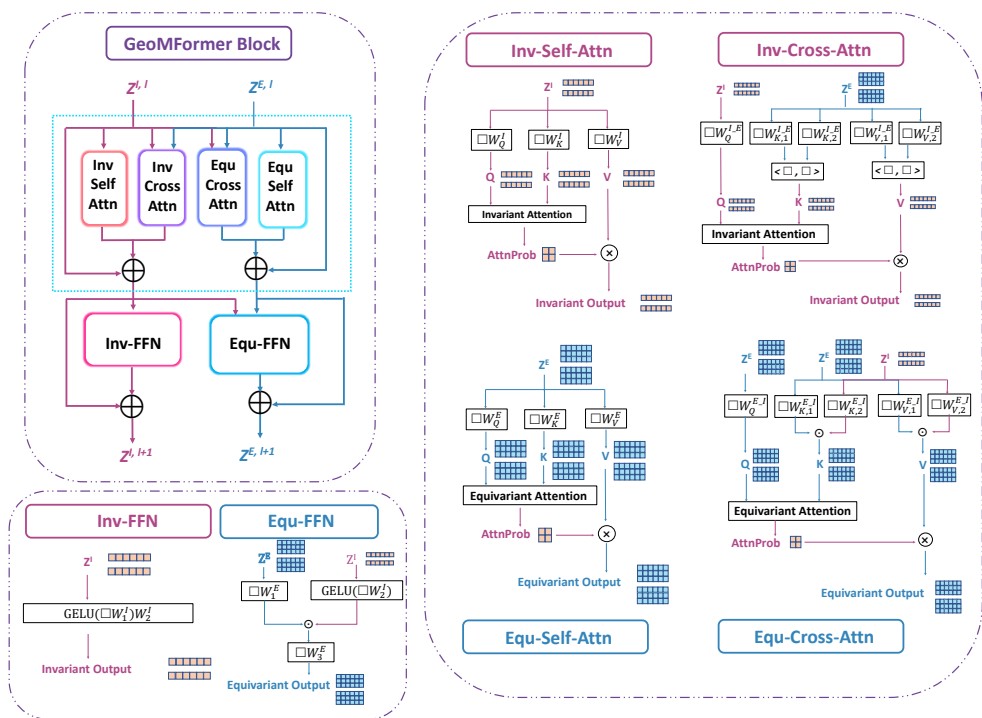

Figure 1: An illustration of our GeoMFormer model architecture.

cross-attention and FFN modules below and prove the properties they satisfy in the appendix. We also incorporate widely used modules like Layer Normalization (Ba et al., 2016) and Structural Encodings (Shi et al., 2022) for better empirical performance. Due to the space limits, we refer readers to the appendix for further details.

**Instantiation of Self-Attention.** In GeoMFormer, the linear function is used to implement both $\psi^I : \mathbb{R}^{n \times d} \to \mathbb{R}^{n \times d}$ and $\psi^E : \mathbb{R}^{n \times 3 \times d} \to \mathbb{R}^{n \times 3 \times d}$:

$$\begin{aligned} \mathbf{Q}^I = \psi_Q^I(\mathbf{Z}^I) = \mathbf{Z}^I W_Q^I, \quad \mathbf{K}^I = \psi_K^I(\mathbf{Z}^I) = \mathbf{Z}^I W_K^I, \quad \mathbf{V}^I = \psi_V^I(\mathbf{Z}^I) = \mathbf{Z}^I W_V^I \\ \mathbf{Q}^E = \psi_Q^E(\mathbf{Z}^E) = \mathbf{Z}^E W_Q^E, \quad \mathbf{K}^E = \psi_K^E(\mathbf{Z}^E) = \mathbf{Z}^E W_K^E, \quad \mathbf{V}^E = \psi_V^E(\mathbf{Z}^E) = \mathbf{Z}^E W_V^E \end{aligned} \tag{6}$$

where $W_Q^I, W_K^I, W_V^I, W_Q^E, W_K^E, W_V^E \in \mathbb{R}^{d \times d_H}$ are learnable parameters.

**Instantiation of Cross-Attention.** As previously stated, both $\psi^{I\text{-}E}$ and $\psi^{E\text{-}I}$ in the cross-attention modules fuse representations from different spaces (invariant & equivariant) into target spaces. In the *Invariant-cross-Equivariant* attention module (Inv-Cross-Attn), to obtain the Key-Value pairs, the equivariant representations are mapped to the invariant space. For the sake of simplicity, we use the dot-product operation $< \cdot, \cdot >$ to instantiate $\psi^{I\text{-}E}$. Given $X, Y \in \mathbb{R}^{n \times 3 \times d}, Z =< X, Y >\in \mathbb{R}^{n \times d}$, where $Z_{[i,k]} = X_{[i,:,k]}^\top Y_{[i,:,k]}$. Then the Key-Value pairs in Inv-Cross-Attn are calculated as:

$$\mathbf{K}^{I\text{-}E} = \psi_K^{I\text{-}E}(\mathbf{Z}^I, \mathbf{Z}^E) =< \mathbf{Z}^E W_{K,1}^{I\text{-}E}, \mathbf{Z}^E W_{K,2}^{I\text{-}E} >, \quad \mathbf{V}^{I\text{-}E} = \psi_V^{I\text{-}E}(\mathbf{Z}^I, \mathbf{Z}^E) =< \mathbf{Z}^E W_{V,1}^{I\text{-}E}, \mathbf{Z}^E W_{V,2}^{I\text{-}E} > \tag{7}$$

where $W_{K,1}^{I\text{-}E}, W_{K,2}^{I\text{-}E}, W_{V,1}^{I\text{-}E}, W_{V,2}^{I\text{-}E} \in \mathbb{R}^{d \times d_H}$ for Key and Value are learnable parameters. On the other hand, the invariant representations are mapped to the equivariant space in the *Equivariant-cross-Invariant* attention module (Equ-Cross-Attn). To achieve this goal, we use the scalar product $\odot$ to instantiate $\psi^{E\text{-}I}$. Given $X \in \mathbb{R}^{n \times 3 \times d}, Y \in \mathbb{R}^{n \times d}, Z = X \odot Y \in \mathbb{R}^{n \times 3 \times d}$, where $Z_{[i,j,k]} = X_{[i,j,k]} \cdot Y_{[i,k]}$. Using this operation, the Key-Value pairs in Equ-Cross-Attn are calculated as:

$$\mathbf{K}^{E\text{-}I} = \psi_K^{E\text{-}I}(\mathbf{Z}^E, \mathbf{Z}^I) = \mathbf{Z}^E W_{K,1}^{E\text{-}I} \odot \mathbf{Z}^I W_{K,2}^{E\text{-}I}, \quad \mathbf{V}^{E\text{-}I} = \psi_V^{E\text{-}I}(\mathbf{Z}^E, \mathbf{Z}^I) = \mathbf{Z}^E W_{V,1}^{E\text{-}I} \odot \mathbf{Z}^I W_{V,2}^{E\text{-}I} \tag{8}$$

where $W_{K,1}^{E\text{-}I}, W_{K,2}^{E\text{-}I}, W_{V,1}^{E\text{-}I}, W_{V,2}^{E\text{-}I} \in \mathbb{R}^{d \times d_H}$ are learnable parameters.

**Instantiation of Feed-Forward Networks.** Besides the attention modules, the feed-forward networks also play important roles in refining contextual representations. In the invariant stream, the feed-forward network is kept unchanged from the standard Transformer model, i.e., Inv-FFN($\mathbf{Z''}^I$) =

GELU($\mathbf{Z}''^I W_1^I$)$W_2^I$, where $W_1^I \in \mathbb{R}^{d \times r}, W_2^I \in \mathbb{R}^{r \times d}$ and $r$ denotes the hidden dimension of the FFN layer. In the equivariant stream, it is worth noting that commonly used non-linear activation functions break the equivariant constraints. In our GeoMFormer, we use the invariant representations as a gating function to non-linearly activate the equivariant representations, i.e., Equ-FFN($\mathbf{Z}''^E$) = ($\mathbf{Z}''^E W_1^E \odot$ GELU($\mathbf{Z}''^I W_2^I$))$W_3^E$, where $W_1^E, W_1^I \in \mathbb{R}^{d \times r}, W_2^E \in \mathbb{R}^{r \times d}$.

**Input Layer.** Given a molecular system $\mathcal{M} = (\mathbf{X}, R)$, we set the invariant representation at the input as $\mathbf{Z}^{I,0} = \mathbf{X}$, where $\mathbf{X}_i \in \mathbb{R}^d$ is a learnable embedding vector indexed by the atom $i$'s type. For the equivariant representation, we set $\mathbf{Z}_i^{E,0} = \hat{\mathbf{r}}_i' g(||\mathbf{r}_i'||)^\top \in \mathbb{R}^{3 \times d}$, where we consider both the direction $\hat{\mathbf{r}}_i' \in \mathbb{R}^3$ and the scale $g(||\mathbf{r}_i'||) \in \mathbb{R}^d$ of the each atom's mean-centered position $\mathbf{r}_i'$. $g : \mathbb{R} \to \mathbb{R}^d$ is instantiated by the Gaussian Basis Kernel, i.e., $g(||\mathbf{r}_i'||) = \psi_i W$, $\psi_i = [\psi_i^1; ...; \psi_i^d]^\top$, $\psi_i^k = -\frac{1}{\sqrt{2\pi}|\sigma^k|} \exp\left(-\frac{1}{2}\left(\frac{\gamma_i ||\mathbf{r}_i'|| + \beta_i - \mu^k}{|\sigma^k|}\right)^2\right), k = 1, ..., d$, where $W \in \mathbb{R}^{d \times d}$ is learnable, $\gamma_i, \beta_i$ are learnable scalars indexed by the atom type, and $\mu^k, \sigma^k$ are learnable kernel center and scaling factor of the $k$-th Kernel. Note that our GeoMFormer is not restricted to these choices, which can encode additional features if the constraints are satisfied, as discussed in the appendix.

## 5 EXPERIMENTS

In this section, we empirically investigate our GeoMFormer on extensive tasks. In particular, we carefully design five experiments covering different types of tasks (invariant & equivariant), data (simple molecules & adsorbate-catalyst complexes & particle systems), and scales, as shown in Table 1. Due to space limits, we present more results (MD17, Ablation Studies) in Appendix D.

Table 1: Summarization of empirical evaluation setup.

| Dataset | Task Description | Task Type | Data Type | Training set size |
|---|---|---|---|---|
| OC20, IS2RE (Chanussot et al., 2021) | Equilibrium Energy Prediction (Sec 5.1.1) | Invariant | Adsorbate-Catalyst complex | 460,328 |
| OC20, IS2RS (Chanussot et al., 2021) | Equilibrium Structure Prediction (Sec 5.1.2) | Equivariant | Adsorbate-Catalyst complex | 460,328 |
| PCQM4Mv2 (Hu et al., 2021) | HOMO-LUMO Gap Prediction (Sec 5.2) | Invariant | Simple molecule | 3,378,606 |
| Molecule3D (Wang et al., 2022) | HOMO-LUMO Gap Prediction (Sec 5.3) | Invariant | Simple molecule | 2,339,788 |
| N-Body Simulation (Satorras et al., 2021) | Position Prediction (Sec 5.4) | Equivariant | Particle System | 3,000 |
| MD17 (Chmiela et al., 2017) | Force Field Modeling (Sec D.6) | Inv/Equ | Simple molecule | 950 |

### 5.1 OC20 PERFORMANCE (INVARIANT & EQUIVARIANT)

The Open Catalyst 2020 (OC20) dataset (Chanussot et al., 2021) was created for catalyst discovery and optimization, which has great significance to advance renewable energy processes for crucial social and energy challenges. Each data is in the form of the adsorbate-catalyst complex. Given the initial structure of a complex, Density Functional Theory (DFT) tools are used to accurately simulate the relaxation process until achieving equilibrium. In practical scenarios, the relaxed energy and structure of the complex are of great interest for catalyst discovery. In this regard, we focus on two significant tasks: Initial Structure to Relaxed Energy (IS2RE) and Initial Structure to Relaxed Structure (IS2RS), which require a model to directly predict the relaxed energy and structure given the initial structure as input respectively[1]. The training set for both tasks is composed of over 460,328 catalyst-adsorbate complexes. To better evaluate the model's performance, the validation and test sets consider the in-distribution (ID) and out-of-distribution settings which uses unseen adsorbates (OOD-Ads), catalysts (OOD-Cat) or both (OOD-Both), containing approximately 200,000 complexes in total.

#### 5.1.1 IS2RE PERFORMANCE (INVARIANT)

As an energy prediction task, the IS2RE task evaluates how well the model learns invariant representations. We follow the experimental setup of Graphormer-3D (Shi et al., 2022). The metric of the IS2RE task is the Mean Absolute Error (MAE) and the percentage of data instances in which the predicted energy is within a 0.02 eV threshold (EwT). We choose several strong baselines covering geometric molecular models using different approaches. Due to space limits, the detailed description of training settings and baselines is presented in the appendix. The results are shown in Table 2. Our

---

[1]Instead of using the iterative relaxation setting that requires massive single-point structure-to-energy-force data to training a force-field model (Chanussot et al., 2021), here we focus on the direct prediction setting that only uses initial-relaxed structure pairs data as the input and label, which is efficient while more challenging.

Table 2: Results on OC20 IS2RE validation set. We report the official results of baselines from the original paper. Bold values indicate the best performance.

| | Energy MAE (eV) ↓ | | | | | EwT (%) ↑ | | | | |
|---|---|---|---|---|---|---|---|---|---|---|
| Model | ID | OOD Ads. | OOD Cat. | OOD Both | Average | ID | OOD Ads. | OOD Cat. | OOD Both | Average |
| CGCNN (Xie & Grossman, 2018) | 0.6203 | 0.7426 | 0.6001 | 0.6708 | 0.6585 | 3.36 | 2.11 | 3.53 | 2.29 | 2.82 |
| SchNet (Schütt et al., 2018) | 0.6465 | 0.7074 | 0.6475 | 0.6626 | 0.6660 | 2.96 | 2.22 | 3.03 | 2.38 | 2.65 |
| DimeNet++ (Gasteiger et al., 2020a) | 0.5636 | 0.7127 | 0.5612 | 0.6492 | 0.6217 | 4.25 | 2.48 | 4.4 | 2.56 | 3.42 |
| GemNet-T (Gasteiger et al., 2021) | 0.5561 | 0.7342 | 0.5659 | 0.6964 | 0.6382 | 4.51 | 2.24 | 4.37 | 2.38 | 3.38 |
| SphereNet (Liu et al., 2022b) | 0.5632 | 0.6682 | 0.5590 | 0.6190 | 0.6024 | 4.56 | 2.70 | 4.59 | 2.70 | 3.64 |
| Graphormer-3D (Shi et al., 2022) | 0.4329 | 0.5850 | 0.4441 | 0.5299 | 0.4980 | - | - | - | - | - |
| GNS (Pfaff et al., 2020) | 0.47 | 0.51 | 0.48 | 0.46 | 0.4800 | - | - | - | - | - |
| Equiformer (Liao & Smidt, 2022) | 0.4156 | 0.4976 | 0.4165 | 0.4344 | 0.4410 | 7.47 | 4.64 | 7.19 | 4.84 | 6.04 |
| GeoMFormer (ours) | **0.3883** | **0.4562** | **0.4037** | **0.4083** | **0.4141** | **11.26** | **6.70** | **9.97** | **6.42** | **8.59** |

GeoMFormer outperforms the compared baselines significantly, achieving impressive performance especially on the out-of-distribution validation sets. In particular, the improvement on the Energy within Threshold (EwT) metric is also significant considering the challenging task. The results indeed demonstrate the effectiveness of our GeoMFormer on learning invariant representations.

### 5.1.2 IS2RS PERFORMANCE (EQUIVARIANT)

Furthermore, we use the IS2RS task to evaluate the model's ability to perform the equivariant prediction task. The metric of the IS2RS task is the Average Distance within Threshold (ADwT) across different thresholds. The Distance within Threshold is computed as the percentage of structures with the atom position MAE below the threshold. We re-implement several competitive baselines under the direct prediction setting for comparison. We refer the readers to the appendix for more details on the

Table 3: Results on OC20 IS2RS validation set. All models are trained and evaluated under the direct prediction setting. Bold values indicate the best.

| | ADwT (%) ↑ | | | | |
|---|---|---|---|---|---|
| Model | ID | OOD Ads | OOD Cat | OOD Both | Average |
| PaiNN (Schütt et al., 2021) | 3.29 | 2.37 | 3.10 | 2.33 | 2.77 |
| TorchMD-Net (Thölke & De Fabritiis, 2022) | 3.32 | 3.35 | 2.94 | 2.89 | 3.13 |
| Spinconv (Shuaibi et al., 2021) | 5.81 | 4.88 | 5.63 | 4.84 | 5.29 |
| GemNet-dT (Gasteiger et al., 2021) | 6.87 | 7.10 | 6.03 | 7.08 | 6.77 |
| GemNet-OC (Gasteiger et al., 2022) | 11.31 | **12.20** | 4.40 | 5.55 | 8.36 |
| GeoMFormer (ours) | **11.45** | 10.52 | **9.94** | **10.78** | **10.67** |

settings. From Table 3, we can see that the IS2RS task under the direct prediction setting is rather difficult. The compared baseline models consistently achieve low ADwT. Our GeoMFormer achieves the best, which indeed verifies the superior ability of our model to perform equivariant molecular tasks.

### 5.2 PCQM4Mv2 PERFORMANCE (INVARIANT)

PCQM4Mv2 is one of the largest quantum chemical property datasets from the OGB Large-Scale Challenge (OGB-LSC (Hu et al., 2021)). Given a molecule, its HOMO-LUMO energy gap of the equilibrium structure is required to predict, which evaluates the model's ability of invariant prediction. This property is highly related to reactivity, photoexcitation, charge transport, and other real applications. DFT tools are used to calculate the HOMO-LUMO gap for ground-truth labels. The total number of training samples is around 3.37 million.

In a practical setting, the DFT-calculated equilibrium geometric structure of each training sample is provided, while only initial structures can be generated by efficient but inaccurate tools (e.g.,

Table 4: Results on PCQM4Mv2. The evaluation metric is the Mean Absolute Error (MAE). We report the official results of baselines. ∗ indicates the best performance achieved by models with the same complexity ($n$ denotes the number of atoms).

| Model | Complexity | # param. | Valid MAE ↓ |
|---|---|---|---|
| MLP-Fingerprint (Hu et al., 2021) | | 16.1M | 0.1735 |
| GINE-VN (Brossard et al., 2020; Gilmer et al., 2017) | | 13.2M | 0.1167 |
| GCN-VN (Kipf & Welling, 2016; Gilmer et al., 2017) | $\mathcal{O}(n)$ | 4.9M | 0.1153 |
| GIN-VN (Xu et al., 2019; Gilmer et al., 2017) | | 6.7M | 0.1083 |
| DeeperGCN-VN (Li et al., 2020; Gilmer et al., 2017) | | 25.5M | 0.1021* |
| TokenGT (Kim et al., 2022) | | 48.5M | 0.0910 |
| EGT (Hussain et al., 2022) | | 89.3M | 0.0869 |
| GRPE (Park et al., 2022) | | 46.2M | 0.0867 |
| Graphormer (Ying et al., 2021a; Shi et al., 2022) | $\mathcal{O}(n^2)$ | 47.1M | 0.0864 |
| GraphGPS (Rampvsek et al., 2022) | | 19.4M | 0.0858 |
| GPS++ (Masters et al., 2022) | | 44.3M | 0.0778 |
| Transformer-M (Luo et al., 2022) | | 47.1M | 0.0787 |
| GEM-2 (Liu et al., 2022a) | $\mathcal{O}(n^3)$ | 32.1M | 0.0793 |
| Uni-Mol+ (Lu et al., 2023) | | 52.4M | 0.0708* |
| GeoMFormer (ours) | $\mathcal{O}(n^2)$ | 54.5M | 0.0734* |

RDKit (Landrum, 2016)) for each validation sample. In this regard, we adopt one recent approach (Uni-Mol+ (Lu et al., 2023)) to handle this task. During training, the model receives RDKit-generated initial structures as the input, and predicts both the HOMO-LUMO energy gap and the equilibrium structure by using both invariant and equivariant representations. After training, the model can be used to predict the HOMO-LUMO gap target by only using the initial structure, which meets the requirement of the settings. We compare various baselines in the leaderboard for comparison. More details of the settings are presented in the appendix.

Table 5: Results on Molecule3D for both random and scaffold splits. We report the official results of baselines. Bold values indicate the best.

| Model | MAE ↓ | |
| --- | --- | --- |
| | Random | Scaffold |
| GIN-Virtual (Hu et al., 2021) | 0.1036 | 0.2371 |
| SchNet (Schütt et al., 2018) | 0.0428 | 0.1511 |
| DimeNet++ (Gasteiger et al., 2020a) | 0.0306 | 0.1214 |
| SphereNet (Liu et al., 2022b) | 0.0301 | 0.1182 |
| ComENet (Wang et al., 2022) | 0.0326 | 0.1273 |
| PaiNN (Schütt et al., 2021) | 0.0311 | 0.1208 |
| TorchMD-Net (Thölke & De Fabritiis, 2022) | 0.0303 | 0.1196 |
| GeoMFormer (ours) | **0.0252** | **0.1045** |

Table 6: Results on N-body System Simulation experiment. We report the official results of baselines. Bold values indicate the best.

| Model | MSE ↓ |
| --- | --- |
| SE(3) Transformer (Fuchs et al., 2020) | 0.0244 |
| Tensor Field Network (Thomas et al., 2018) | 0.0155 |
| Graph Neural Network (Gilmer et al., 2017) | 0.0107 |
| Radial Field (Köhler et al., 2019) | 0.0104 |
| EGNN (Satorras et al., 2021) | 0.0071 |
| GeoMFormer(ours) | **0.0047** |

From Table 4. Our GeoMFormer achieves the lowest MAE among the quadratic models, e.g., 6.7% relative MAE reduction compared to the previous best model. Besides, compared to the best model Uni-Mol+ (Lu et al., 2023), our GeoMFormer achieves competitive performance while keeping the efficiency ($\mathcal{O}(n^2)$ complexity), which can be more broadly applied to large molecular systems. Overall, the results further verify the effectiveness of GeoMFormer on invariant representation learning.

### 5.3 MOLECULE3D PERFORMANCE (INVARIANT)

Molecule3D (Xu et al., 2021) is a newly proposed large-scale dataset curated from the PubChemQC project (Maho, 2015; Nakata & Shimazaki, 2017). Each molecule has the DFT-calculated equilibrium geometric structure. The task is to predict the HOMO-LUMO energy gap, which is the same as PCQM4Mv2. The dataset contains 3,899,647 molecules in total and is split into training, validation, and test sets with the splitting ratio $6 : 2 : 2$. In particular, both random and scaffold splitting methods are adopted to thoroughly evaluate the in-distribution and out-of-distribution performance of geometric molecular models. Following (Wang et al., 2022), we compare our GeoMFormer with several competitive baselines. Detailed descriptions of the training settings and baselines are presented in the appendix. It can be easily seen from Table 5 that our GeoMFormer consistently outperforms all baselines on both random and scaffold split settings, e.g., 16.3% and 11.6% relative MAE reduction compared to the previous best model respectively.

### 5.4 N-BODY SIMULATION PERFORMANCE (EQUIVARIANT)

Simulating dynamical systems consisting of a set of geometric objects interacting under physical laws is crucial in many applications, e.g. molecular dynamic simulation. Following (Fuchs et al., 2020; Satorras et al., 2021), we use a synthetic n-body system simulation task as an extension of molecular modeling tasks. This task requires the model to forecast the positions of a set of particles, which are modeled by simple interaction rules, yet can exhibit complex dynamics. Thus, the model's ability to perform equivariant prediction tasks is carefully evaluated. In this dataset, the simulated system consists of 5 particles, each of which carries a positive or negative charge and has an initial position and velocity in the three-dimensional Euclidean space. The system is controlled by physical rules involving attractive and repulsive forces. The dataset contains 3.000 trajectories for training, 2.000 trajectories for validation, and 2.000 trajectories for testing. We compare several competitive baselines following (Satorras et al., 2021). Due to space limits, the detailed descriptions of the data generation, training settings and baselines are presented in the appendix. The results are shown in Table 6. Our GeoMFormer achieves the best performance compared to all baselines. In particular, the significant 33.8% MSE reduction indeed demonstrates the GeoMFormer's superior ability on learning equivariant representations.

## 6 CONCLUSION

In this paper, we propose a general and flexible architecture, called GeoMFormer, for learning geometric molecular representations. Using the standard Transformer backbone, two streams are developed for learning invariant and equivariant representations respectively. In particular, the cross-attention mechanism is used to bridge these two streams, letting each stream leverage contextual information from the other stream and enhance its representations. This simple yet effective design significantly boosts both invariant and equivariant modeling. Within the newly proposed framework, many existing methods can be regarded as special instances, showing the generality of our method. We conduct extensive experiments covering diverse tasks, data and scales. All the empirical results show that our GeoMFormer can achieve strong performance in different scenarios. The potential of our GeoMFormer can be further explored in a broad range of applications in molecular modeling.

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

# A   IMPLEMENTATION DETAILS OF GEOMFORMER

**Layer Normalizations.**   Being a Transformer-based model, GeoMFormer also adopts the layer normalization (LN) (Ba et al., 2016) module for training stability. In the invariant stream, the LN module remains unchanged from the standard design (Ba et al., 2016; Xiong et al., 2020). In particular, we specialized the LN module as Equ-LN in the equivariant stream to satisfy the geometric constraints. Formally, given the equivariant representation $\mathbf{z}_i^E \in \mathbb{R}^{3\times d}$ of the atom $i$, Equ-LN$(\mathbf{z}_i^E) = \mathbf{U}(\mathbf{z}_i^E - \mu\mathbf{1}^\top) \odot \gamma$, where $\mu = \frac{1}{d}\sum_{k=1}^d \mathbf{Z}_{[i,:,k]}^E \in \mathbb{R}^3$, $\gamma \in \mathbb{R}^d$ is a learnable vector, and $\mathbf{U} \in \mathbb{R}^{3\times 3}$ denotes the inverse square root of the covariance matrix, i.e., $\mathbf{U}^{-2} = \frac{(\mathbf{z}_i^E - \mu\mathbf{1}^\top)(\mathbf{z}_i^E - \mu\mathbf{1}^\top)^\top}{\mathbf{d}}$.

**Structural Encodings.**   We follow (Shi et al., 2022) to incorporate the 3D structural encoding, which serves as the bias term in the softmax attention module. In particular, we consider the Euclidean distance $||\mathbf{r}_i - \mathbf{r}_j||$ between atom $i$ and $j$. The Gaussian Basis Kernel function (Scholkopf et al., 1997) is used to encode the interatomic distance, i.e., $b_{(i,j)}^k = -\frac{1}{\sqrt{2\pi}|\sigma^k|}\exp(-\frac{1}{2}(\frac{\gamma_{(i,j)}||\mathbf{r}_i-\mathbf{r}_j||+\beta_{(i,j)}-\mu^k}{|\sigma^k|})^2), k = 1,...,K$, where $K$ is the number of Gaussian Basis kernels. The 3D structural encoding is obtained by $B_{ij} = \text{GELU}(b_{(i,j)}W_D^1)W_D^2$, where $b_{(i,j)} = [b_{(i,j)}^1;...;b_{(i,j)}^K]^\top, W_D^1 \in \mathbb{R}^{K\times K}, W_D^2 \in \mathbb{R}^{K\times 1}$ are learnable parameters. $\gamma_{(i,j)}, \beta_{(i,j)}$ are learnable scalars indexed by the pair of atom types, and $\mu^k, \sigma^k$ are learnable kernel center and learnable scaling factor of the $k$-th Gaussian Basis Kernel. Denote $B$ as the matrix form of the 3D distance encoding, whose shape is $n \times n$. Then the attention probability is calculated by softmax$(\frac{QK^\top}{\sqrt{d}} + B)$, where $Q$ and $K$ are the query and key introduced in Section 3.

# B   PROOF OF GEOMETRIC CONSTRAINTS

In this section, we provide thorough proof of the aforementioned conditions in Section 4 that satisfy the geometric constraints. For the sake of convenience, we restate the notations and geometric constraints here. Formally, let $V_{\mathcal{M}}$ denote the space of molecular systems, for each atom $i$, we define equivariant representation $\phi^E$ and invariant representation $\phi^I$ if $\forall g = (\mathbf{t}, \mathbf{R}) \in SE(3), \mathcal{M} = (\mathbf{X}, R) \in V_{\mathcal{M}}$, the following conditions are satisfied:

$$\phi^E : V_{\mathcal{M}} \to \mathbb{R}^{3\times d}, \qquad \mathbf{R}\phi^E(\mathbf{X}, \{\mathbf{r}_1, ..., \mathbf{r}_n\}) = \phi^E(\mathbf{X}, \{\mathbf{R}\mathbf{r}_1, ..., \mathbf{R}\mathbf{r}_n\}) \qquad (9a)$$

$$\phi^E : V_{\mathcal{M}} \to \mathbb{R}^{3\times d}, \qquad \phi^E(\mathbf{X}, \{\mathbf{r}_1, ..., \mathbf{r}_n\}) = \phi^E(\mathbf{X}, \{\mathbf{r}_1 + \mathbf{t}, ..., \mathbf{r}_n + \mathbf{t}\}) \qquad (9b)$$

$$\phi^I : V_{\mathcal{M}} \to \mathbb{R}^d, \qquad \phi^I(\mathbf{X}, \{\mathbf{r}_1, ..., \mathbf{r}_n\}) = \phi^I(\mathbf{X}, \{\mathbf{R}\mathbf{r}_1 + \mathbf{t}, ..., \mathbf{R}\mathbf{r}_n + \mathbf{t}\}) \qquad (9c)$$

where $\mathbf{t} \in \mathbb{R}^3, \mathbf{R} \in \mathbb{R}^{3\times 3}, \det(\mathbf{R}) = 1$ and $\mathbf{X} \in \mathbb{R}^{n\times d}$ denotes the atoms with features, $R = \{\mathbf{r}_1, ..., \mathbf{r}_n\}, \mathbf{r}_i \in \mathbb{R}^3$ denotes the cartesian coordinate of atom $i$. We present the proof of the General Design Philosophy (Section B.1) and our GeoMFormer model (Section B.2) respectively.

## B.1   PROOF OF THE GENERAL DESIGN PHILOSOPHY.

Given invariant and equivariant representations $\mathbf{Z}^{I,0} \in \mathbb{R}^{n\times d}, \mathbf{Z}^{E,0} \in \mathbb{R}^{n\times 3\times d}$ at the input, we prove that the update rules shown in Eqn.(4) satisfy the above constraints in proper conditions. In particular, we first separately study each component of the block, i.e., Inv-Self-Attn, Equ-Self-Attn, Inv-Cross-Attn, Equ-Cross-Attn, and then check the properties of the whole framework.

**Invariant Self-Attention.**   Given invariant representation $\mathbf{Z}^{I,l} \in \mathbb{R}^{n\times d}, \mathbf{Q}^{I,l} = \psi_Q^{I,l}(\mathbf{Z}^{I,l}), \mathbf{K}^{I,l} = \psi_K^{I,l}(\mathbf{Z}^{I,l}), \mathbf{V}^{I,l} = \psi_V^{I,l}(\mathbf{Z}^{I,l})$, as stated in Section 4.1, where $\psi^{I,l} : \mathbb{R}^{n\times d} \to \mathbb{R}^{n\times d}$ is invariant. In this regard, $\forall g = (\mathbf{t}, \mathbf{R}) \in SE(3), \mathbf{Q}^{I,l}, \mathbf{K}^{I,l}, \mathbf{V}^{I,l}$ remain unchanged, which means that Inv-Self-Attn$(\mathbf{Q}^{I,l}, \mathbf{K}^{I,l}, \mathbf{V}^{I,l})$ also remains unchanged. Then the invariance of the output representations is preserved.

**Equivariant Self-Attention.** Given equivariant representation $\mathbf{Z}^{E,l} \in \mathbb{R}^{n\times 3\times d}$, $\mathbf{Q}^{E,l} = \psi_Q^{E,l}(\mathbf{Z}^{E,l})$, $\mathbf{K}^{E,l} = \psi_K^{E,l}(\mathbf{Z}^{E,l})$, $\mathbf{V}^{E,l} = \psi_V^{E,l}(\mathbf{Z}^{E,l})$, as stated in Section 4.1, where $\psi^{E,l} : \mathbb{R}^{n\times 3\times d} \to \mathbb{R}^{n\times 3\times d}$ is equivariant. Besides, the attention score is modified as $\alpha_{ij} = \sum_{k=1}^{d} \mathbf{Q}^E_{[i,:,k]} {\mathbf{K}^E_{[j,:,k]}}^{\top}$, where $\mathbf{Q}^E_{[i,:,k]} \in \mathbb{R}^3$ denotes the $k$-th dimension of the atom $i$'s Query. First, we check the rotation equivariance of the Equ-Self-Attn. Given any orthogonal transformation matrix $\mathbf{R} \in \mathbb{R}^{3\times 3}$, $\det(\mathbf{R}) = 1$, we have $\sum_{k=1}^{d} \mathbf{Q}^E_{[i,:,k]} \mathbf{R}(\mathbf{K}^E_{[j,:,k]}\mathbf{R})^{\top} = \sum_{k=1}^{d} \mathbf{Q}^E_{[i,:,k]} \mathbf{R}\mathbf{R}^{\top} {\mathbf{K}^E_{[j,:,k]}}^{\top} = \sum_{k=1}^{d} \mathbf{Q}^E_{[i,:,k]} {\mathbf{K}^E_{[j,:,k]}}^{\top} = \alpha_{ij}$, which preserves the invariance. As $\psi^{E,l}$ is equivariant, we have $\psi^{E,l}([\mathbf{R}\mathbf{Z}^{E,l}_1; , ..., ; \mathbf{R}\mathbf{Z}^{E,l}_n]) = [\mathbf{R}\psi^{E,l}(\mathbf{Z}^{E,l})_1; , ..., ; \mathbf{R}\psi^{E,l}(\mathbf{Z}^{E,l})_n]$. Since the output equivariant representation of atom $i$ preserves the equivariance, i.e., $\sum_{j=1}^{n} \frac{\exp(\alpha_{ij})}{\sum_{j'=1}^{n}\exp(\alpha_{ij'})} \mathbf{R}\mathbf{V}^{E,l}_j = \mathbf{R}(\sum_{j=1}^{n} \frac{\exp(\alpha_{ij})}{\sum_{j'=1}^{n}\exp(\alpha_{ij'})} \mathbf{V}^{E,l}_j)$, the rotation equivariance is satisfied. Moreover, since the equivariant representation $\mathbf{Z}^{E,l}$ preserves the translation invariance (Eqn.(9b)), the output equivariant representation of Equ-Self-Attn naturally satisfies this constraint.

**Cross-Attention modules.** As stated in Section 4.1, the Query, Key, and Value of Inv-Cross-Attn are specified as $\mathbf{Q}^{I\_E,l} = \psi_Q^{I,l}(\mathbf{Z}^{I,l})$, $\mathbf{K}^{I\_E,l} = \psi_K^{I\_E,l}(\mathbf{Z}^{I,l}, \mathbf{Z}^{E,l})$, $\mathbf{V}^{I\_E,l} = \psi_V^{I\_E,l}(\mathbf{Z}^{I,l}, \mathbf{Z}^{E,l})$, where $\psi^{I,l}, \psi^{I\_E,l}$ are invariant. That is to say, $\forall g = (\mathbf{t}, \mathbf{R}) \in SE(3)$, $\mathbf{Q}^{I\_E,l}, \mathbf{K}^{I\_E,l}, \mathbf{V}^{I\_E,l}$ remain unchanged. Then the invariance of its output representations is preserved as in Inv-Self-Attn. On the other hand, the Query, Key, and Value of Equ-Cross-Attn are specified as $\mathbf{Q}^{E\_I,l} = \psi_Q^{E,l}(\mathbf{Z}^{E,l})$, $\mathbf{K}^{E\_I,l} = \psi_K^{E\_I,l}(\mathbf{Z}^{E,l}, \mathbf{Z}^{I,l})$, $\mathbf{V}^{E\_I,l} = \psi_V^{E\_I,l}(\mathbf{Z}^{E,l}, \mathbf{Z}^{I,l})$, where $\psi^{E,l}, \psi^{E\_I,l}$ are equivariant, i.e., $\psi^{E\_I,l}([\mathbf{R}\mathbf{Z}^{E,l}_1; , ..., ; \mathbf{R}\mathbf{Z}^{E,l}_n], \mathbf{Z}^{I,l}) = [\mathbf{R}\psi^{E\_I,l}(\mathbf{Z}^{E,l}, \mathbf{Z}^{I,l})_1; , ..., ; \mathbf{R}\psi^{E,l}(\mathbf{Z}^{E,l}, \mathbf{Z}^{I,l})_n]$ and $\psi^{E,l}([\mathbf{R}\mathbf{Z}^{E,l}_1; , ..., ; \mathbf{R}\mathbf{Z}^{E,l}_n]) = [\mathbf{R}\psi^{E,l}(\mathbf{Z}^{E,l})_1; , ..., ; \mathbf{R}\psi^{E,l}(\mathbf{Z}^{E,l})_n]$. As stated in Equ-Self-Attn, the output equivariant representations of Equ-Cross-Attn preserve the rotation equivariance. Similarly, the translation invariance property is also naturally satisfied.

**Feed-Forward Networks.** As Inv-FFN and Equ-FFN satisfy the invariance and equivariance constraints respectively, we can directly obtain that $\forall g = (\mathbf{t}, \mathbf{R}) \in SE(3)$, the output of Inv-FFN remains unchanged, and the output of Equ-FFN preserves the rotation equivariance, i.e., $\text{Equ-FFN}([\mathbf{R}\mathbf{Z}^{E,l}_1; , ..., ; \mathbf{R}\mathbf{Z}^{E,l}_n]) = [\mathbf{R}\,\text{Equ-FFN}(\mathbf{Z}^{E,l})_1; , ..., ; \mathbf{R}\,\text{Equ-FFN}(\mathbf{Z}^{E,l})_n]$. The translation invariance is also naturally preserved by Equ-FFN.

With the above analysis, the update rules stated in Eqn.(4) satisfy the geometric constraints (Eqn.(9a), Eqn.(9b) and Eqn.(9c)). As our model is composed of stacked blocks, the invariant and equivariant output representations of the whole model also preserve the constraints.

### B.2 PROOF OF THE GEOMFORMER

Next, we provide proof of the instantiation of our GeoMFormer in Section 4.2 that satisfies the geometric constraints. Similarly, we separately check the properties of each component as our GeoMFormer is composed of stacked GeoMFormer blocks. Once the constraints are satisfied by each component, the output invariant and equivariant representations of the whole model naturally satisfy the geometric constraints (Eqn.(9a), Eqn.(9b) and Eqn.(9c)).

**Input layer.** As stated in Section 4.2, the invariant representation at the input is set as $\mathbf{Z}^{I,0} = \mathbf{X}$, where $\mathbf{X}_i \in \mathbb{R}^d$ is a learnable embedding vector indexed by the atom $i$'s type. Since $\mathbf{Z}^{I,0}$ does not contain any information from $R = \{\mathbf{r}_1, ..., \mathbf{r}_n\}$, it naturally satisifies the invariance constraint (Eqn.(9c)). The equivariant representation at the input is set as $\mathbf{Z}^{E,0}_i = \hat{\mathbf{r}}'_i g(||\mathbf{r}'_i||)^{\top} \in \mathbb{R}^{3\times d}$, where $\mathbf{r}'_i$ denotes the mean-centered position of atom $i$, i.e., $\mathbf{r}'_i = \mathbf{r}_i - \frac{1}{n}\sum_{k=1}^{n} \mathbf{r}_k$, $\hat{\mathbf{r}}'_i = \frac{\mathbf{r}'_i}{||\mathbf{r}'_i||}$ , and $g : \mathbb{R} \to \mathbb{R}^d$ is instantiated by the Gaussian Basis Kernel function. First, the translation invariance constraint (Eqn.(9b)) is satisfied. Given any translation vector $\mathbf{t} \in \mathbb{R}^3$, $\mathbf{r}_i + \mathbf{t} - \frac{1}{n}\sum_{k=1}^{n}(\mathbf{r}_k + \mathbf{t}) = \mathbf{r}_i - \frac{1}{n}\sum_{k=1}^{n} \mathbf{r}_k$, and $\mathbf{Z}^{E,0}_i$ remains unchanged. Second, the rotation equivariance (Eqn.(9a)) is also preserved. Given any orthogonal transformation matrix $\mathbf{R} \in \mathbb{R}^{3\times 3}$, $\det(\mathbf{R}) = 1$, we have

$||\mathbf{R}\mathbf{r}_i'|| = ||\mathbf{r}_i'||$. With $\mathbf{R}\mathbf{r}_i$ as the input, we have $\mathbf{R}\mathbf{r}_i - \frac{1}{n}\sum_{k=1}^{n}\mathbf{R}\mathbf{r}_k = \mathbf{R}(\mathbf{r}_i - \frac{1}{n}\sum_{k=1}^{n}\mathbf{r}_k) = \mathbf{R}\mathbf{r}_i'$ and $g(||\mathbf{R}\mathbf{r}_i'||) = g(||\mathbf{r}_i'||)$, which means that the rotation equivariance constraint is satisfied.

**Self-Attention modules.** For Inv-Self-Attn and Equ-Self-Attn, we use the linear function to implement both $\psi^I$ and $\psi^E$, i.e., $\mathbf{Q}^I = \psi_Q^I(\mathbf{Z}^I) = \mathbf{Z}^I W_Q^I, \mathbf{K}^I = \psi_K^I(\mathbf{Z}^I) = \mathbf{Z}^I W_K^I, \mathbf{V}^I = \psi_V^I(\mathbf{Z}^I) = \mathbf{Z}^I W_V^I$ and $\mathbf{Q}^E = \psi_Q^E(\mathbf{Z}^E) = \mathbf{Z}^E W_Q^E, \mathbf{K}^E = \psi_K^E(\mathbf{Z}^E) = \mathbf{Z}^E W_K^E, \mathbf{V}^E = \psi_V^E(\mathbf{Z}^E) = \mathbf{Z}^E W_V^E$. It is straightforward that the conditions mentioned in Section B.1 are satisfied. The linear function keeps the invariance of $\mathbf{Z}^I$ (Eqn.(9c)) and the rotation equivariance of $\mathbf{Z}^E$ (Eqn.(9a)), e.g., $\forall \mathbf{R} \in \mathbb{R}^{3\times3}, \det(\mathbf{R}) = 1, (\mathbf{R}\mathbf{Z}_i^E)W_Q^E = \mathbf{R}(\mathbf{Z}_i^E W_Q^E) = \mathbf{R}\mathbf{Z}_i^E$. Note that the translation invariance of $\mathbf{Z}^E$ (Eqn.(9b)) is not changed by the linear function.

**Cross-Attention modules.** For Inv-Cross-Attn, we use the linear function to implement $\psi_Q^I$, which satisfies the constraints as previously stated. Besides, we instantiate $\mathbf{K}^{I\_E}$ and $\mathbf{V}^{I\_E}$ as $\mathbf{K}^{I\_E} = \psi_K^{I\_E}(\mathbf{Z}^I, \mathbf{Z}^E) = < \mathbf{Z}^E W_{K,1}^{I\_E}, \mathbf{Z}^E W_{K,2}^{I\_E} >, \mathbf{V}^{I\_E} = \psi_V^{I\_E}(\mathbf{Z}^I, \mathbf{Z}^E) = < \mathbf{Z}^E W_{V,1}^{I\_E}, \mathbf{Z}^E W_{V,2}^{I\_E} >$. Here we prove that such instantiation preserve the invariance. First, given any orthogonal transformation matrix $\mathbf{R} \in \mathbb{R}^{3\times3}, \det(\mathbf{R}) = 1$, we have $< ([\mathbf{R}\mathbf{Z}_1^E; ...; \mathbf{R}\mathbf{Z}_n^E])W_{K,1}^{I\_E}, ([\mathbf{R}\mathbf{Z}_1^E; ...; \mathbf{R}\mathbf{Z}_n^E])W_{K,2}^{I\_E} > = < \mathbf{Z}^E W_{K,1}^{I\_E}, \mathbf{Z}^E W_{K,2}^{I\_E} >$. The reason is that given $X, Y \in \mathbb{R}^{n\times3\times d}, Z = < X, Y > \in \mathbb{R}^{n\times d}$, where $Z_{[i,k]} = X_{[i,:,k]}{}^\top Y_{[i,:,k]} = X_{[i,:,k]}{}^\top \mathbf{R}^\top \mathbf{R} Y_{[i,:,k]} = (\mathbf{R}X_{[i,:,k]})^\top(\mathbf{R}Y_{[i,:,k]})$. The translation invariance of $\mathbf{Z}^E$ is also preserved.

For Equ-Cross-Attn, we also use the linear function to implement $\psi_Q^E$, which satisfies the constraints as previously stated. Besides, we instantiate $\mathbf{K}^{E\_I}$ and $\mathbf{V}^{E\_I}$ as $\mathbf{K}^{E\_I} = \psi_K^{E\_I}(\mathbf{Z}^E, \mathbf{Z}^I) = \mathbf{Z}^E W_{K,1}^{E\_I} \odot \mathbf{Z}^I W_{K,2}^{E\_I}, \mathbf{V}^{E\_I} = \psi_V^{E\_I}(\mathbf{Z}^E, \mathbf{Z}^I) = \mathbf{Z}^E W_{V,1}^{E\_I} \odot \mathbf{Z}^I W_{V,2}^{E\_I}$. First, given any orthogonal transformation matrix $\mathbf{R} \in \mathbb{R}^{3\times3}$, we have $([\mathbf{R}\mathbf{Z}_1^E; ...; \mathbf{R}\mathbf{Z}_n^E])W_{K,1}^{E\_I} \odot \mathbf{Z}^I W_{K,2}^{E\_I} = [\mathbf{R}(\mathbf{Z}^E W_{K,1}^{E\_I} \odot \mathbf{Z}^I W_{K,2}^{E\_I})_1; ...; \mathbf{R}(\mathbf{Z}^E W_{K,1}^{E\_I} \odot \mathbf{Z}^I W_{K,2}^{E\_I})_n]$, which preserves the rotation equivariance. The reason lies in that given $X \in \mathbb{R}^{n\times3\times d}, Y \in \mathbb{R}^{n\times d}, Z_i = \mathbf{R}X_i \odot Y_i \in \mathbb{R}^{3\times d}$, where $Z_{[i,:,k]} = (\mathbf{R}X_{[i,:,k]}) \cdot Y_{[i,k]} = \mathbf{R}(X_{[i,:,k]} \cdot Y_{[i,k]})$. Additionally, the translation invariance of both $\mathbf{K}^{E\_I}$ and $\mathbf{V}^{E\_I}$ is preserved because of the translation invariance of $\mathbf{Z}^E$ and $\mathbf{Z}^I$. In this way, the instantiations of cross-attention modules satisfy the geometric constraints.

**Feed-Forward Networks.** For Inv-FFN$(\mathbf{Z}''^I) = $ GELU$(\mathbf{Z}''^I W_1^I)W_2^I$, the invariance constraint (Eqn. 9c) is naturally preserved. For Equ-FFN$(\mathbf{Z}''^E) = (\mathbf{Z}''^E W_1^E \odot$ GELU$(\mathbf{Z}''^I W_2^I))W_3^E$, the rotation equivariance constraint is also similarly preserved as in Equ-Cross-Attn. Besides, the translation invariance of Equ-FFN$(\mathbf{Z}''^E)$ is also preserved with the property of $\mathbf{Z}''^E$ and $\mathbf{Z}''^I$.

**Layer Normalizations.** As introduced in Section A, we use the layer normalization modules for both invariant and equivariant streams. For the invariant stream, the layer normalization remains unchanged, and the invariance constraint is naturally preserved. For the equivariant stream, given the equivariant representation $\mathbf{z}_i^E \in \mathbb{R}^{3\times d}$ of the atom $i$, Equ-LN$(\mathbf{z}_i^E) = \mathbf{U}(\mathbf{z}_i^E - \mu\mathbf{1}^\top) \odot \gamma$, where $\mu = \frac{1}{d}\sum_{k=1}^{d}\mathbf{Z}_{[i,:,k]}^E \in \mathbb{R}^3, \gamma \in \mathbb{R}^d$ is a learnable vector, and $\mathbf{U} \in \mathbb{R}^{3\times3}$ denotes the inverse square root of the covariance matrix, i.e., $\mathbf{U}^{-2} = \frac{(\mathbf{z}_i^E - \mu\mathbf{1}^\top)(\mathbf{z}_i^E - \mu\mathbf{1}^\top)^\top}{d}$. First, given any orthogonal transformation matrix $\mathbf{R} \in \mathbb{R}^{3\times3}, \det(\mathbf{R}) = 1, \frac{(\mathbf{R}\mathbf{z}_i^E - \mathbf{R}\mu\mathbf{1}^\top)(\mathbf{R}\mathbf{z}_i^E - \mathbf{R}\mu\mathbf{1}^\top)^\top}{d} = \frac{(\mathbf{R}\mathbf{z}_i^E - \mathbf{R}\mu\mathbf{1}^\top)(\mathbf{R}\mathbf{z}_i^E - \mathbf{R}\mu\mathbf{1}^\top)^\top}{d} = \mathbf{R}\frac{(\mathbf{z}_i^E - \mu\mathbf{1}^\top)(\mathbf{z}_i^E - \mu\mathbf{1}^\top)^\top}{d}\mathbf{R}^\top = \mathbf{R}\mathbf{U}^{-2}\mathbf{R}^\top = \mathbf{R}\mathbf{U}^{-1}\mathbf{R}^\top\mathbf{R}\mathbf{U}^{-1}\mathbf{R}^\top = (\mathbf{R}\mathbf{U}\mathbf{R}^\top)^{-2}$, then we have Equ-LN$(\mathbf{R}\mathbf{z}_i^E) = \mathbf{R}\mathbf{U}\mathbf{R}^\top(\mathbf{R}\mathbf{z}_i^E - \mathbf{R}\mu\mathbf{1}^\top) \odot \gamma = \mathbf{R}(\mathbf{U}(\mathbf{z}_i^E - \mu\mathbf{1}^\top)) = \mathbf{R}$ Equ-LN$(\mathbf{z}_i^E)$, which preserves the rotation equivariance (Eqn.(9a)). The translation invariance of $\mathbf{Z}^E$ is also preserved.

**Structural Encodings.** As introduced in Section A, the structural encodings serve as the bias term in the softmax attention module. Since only the relative distance $||\mathbf{r}_i - \mathbf{r}_j||, \forall i, j \in [n]$ is used, the invariance constraint is preserved, i.e., given $\forall g = (\mathbf{t}, \mathbf{R}) \in SE(3), ||\mathbf{R}\mathbf{r}_i + \mathbf{t} - \mathbf{R}\mathbf{r}_j + \mathbf{t}|| = ||\mathbf{r}_i - \mathbf{r}_j||$.

## C  DISCUSSIONS

### C.1  CONNECTIONS TO PREVIOUS APPROACHES

In this section, we present a detailed discussion of how previous models (PaiNN (Schütt et al., 2021) and TorchMD-Net (Thölke & De Fabritiis, 2022)) can be viewed as special instantiations by extending the design philosophy described in Section 4.1. Without loss of generality, we omit the cutoff conditions used in these works for readability, which can be naturally included in our framework.

**PaiNN (Schütt et al., 2021).** Both invariant representations $\mathbf{Z}^I = [\mathbf{z}_1^{I\top};...;\mathbf{z}_n^{I\top}] \in \mathbb{R}^{n \times d}$ and equivariant representations $\mathbf{Z}^E = [\mathbf{z}_1^E;...;\mathbf{z}_n^E] \in \mathbb{R}^{n \times 3 \times d}$ are maintained in PaiNN, where $\mathbf{z}_i^I \in \mathbb{R}^d$ and $\mathbf{z}_i^E \in \mathbb{R}^{3 \times d}$ are the invariant and equivariant representations for atom $i$, respectively. In each layer, the representations are updated as follows:

$$
\begin{aligned}
\mathbf{Z}'^{I,l} &= \mathbf{Z}^{I,l} + \text{Message-Block-Inv}(\mathbf{Z}^{I,l}) \\
\mathbf{Z}'^{E,l} &= \mathbf{Z}^{E,l} + \text{Message-Block-Equ}(\mathbf{Z}^{I,l}, \mathbf{Z}^{E,l}) \\
\mathbf{Z}^{I,l+1} &= \mathbf{Z}'^{I,l} + \text{Update-Block-Inv}(\mathbf{Z}'^{I,l}, \mathbf{Z}'^{E,l}) \\
\mathbf{Z}^{E,l+1} &= \mathbf{Z}'^{E,l} + \text{Update-Block-Equ}(\mathbf{Z}'^{I,l}, \mathbf{Z}'^{E,l})
\end{aligned}
\tag{10}
$$

In the message block, the invariant and equivariant representations are updated in the following manner. For brevity, we omit the layer index $l$.

$$
\begin{aligned}
\text{Message-Block-Inv}(\mathbf{z}_i^I) &= \sum_j \phi_s(\mathbf{z}_j^I) \circ \mathcal{W}_s(||\mathbf{r}_i - \mathbf{r}_j||) \\
\text{Message-Block-Equ}(\mathbf{z}_i^I, \mathbf{z}_i^E) &= \sum_j \mathbf{z}_j^E \odot \left( \phi_{vv}(\mathbf{z}_j^I) \circ \mathcal{W}_{vv}(||\mathbf{r}_i - \mathbf{r}_j||) \right) \\
&\quad + \frac{\mathbf{r}_i - \mathbf{r}_j}{||\mathbf{r}_i - \mathbf{r}_j||} \left( \phi_{vs}(\mathbf{z}_j^I) \circ \mathcal{W}'_{vs}(||\mathbf{r}_i - \mathbf{r}_j||) \right)^\top
\end{aligned}
\tag{11}
$$

The scalar product $\odot$ is defined the same way as in Section 4.2, i.e., given $x \in \mathbb{R}^{3 \times d}, y \in \mathbb{R}^d, z = x \odot y \in \mathbb{R}^{3 \times d}$, where $z_{[i,j]} = x_{[i,k]} \cdot y_{[k]}$. $\circ$ denotes the element-wise product, $\phi_s, \phi_{vv}, \phi_{vs} : \mathbb{R}^d \to \mathbb{R}^d$ are all 2-layer MLP with the SiLU activation, $\mathcal{W}_s, \mathcal{W}_{vv}, \mathcal{W}'_{vs} : \mathbb{R} \to \mathbb{R}^d$ are instantiated by learnable radial basis functions. $\frac{\mathbf{r}_i - \mathbf{r}_j}{||\mathbf{r}_i - \mathbf{r}_j||} \in \mathbb{R}^3$ denotes the relative direction between atom $i$'s and $j$'s positions.

In the update block, the invariant and equivariant representations are updated in the following manner:

$$
\begin{aligned}
\text{Update-Block-Inv}(\mathbf{z}_i^I, \mathbf{z}_i^E) &= \mathbf{a}_{ss}(\mathbf{z}_i^I, ||\mathbf{z}_i^E \mathbf{V}||) + \mathbf{a}_{sv}(\mathbf{z}_i^I, ||\mathbf{z}_i^E \mathbf{V}||) \circ <\mathbf{z}_i^E \mathbf{U}, \mathbf{z}_i^E \mathbf{V}> \\
\text{Update-Block-Equ}(\mathbf{z}_i^I, \mathbf{z}_i^E) &= \mathbf{a}_{vv}(\mathbf{z}_i^I, ||\mathbf{z}_i^E \mathbf{V}||) \odot (\mathbf{z}_i^E \mathbf{U})
\end{aligned}
$$

$\mathbf{V}, \mathbf{U} \in \mathbb{R}^{d \times d}$ are learnable parameters. $< \cdot, \cdot >$ is defined the same way as in Section 4.2, i.e., given $x, y \in \mathbb{R}^{3 \times d}, z = <x, y> \in \mathbb{R}^d$, where $z_{[k]} = x_{[:,k]}^\top y_{[:,k]}$. Norm $|| \cdot || : \mathbb{R}^{3 \times d} \to \mathbb{R}^d$ is calculated along the spatial dimension, i.e., $|| \cdot || = < \cdot, \cdot >$. $\circ$ denotes the element-wise product. $\odot$ is also defined the same as in Section 4.2. $\mathbf{a}(\cdot, \cdot) : \mathbb{R}^d \times \mathbb{R}^d \to \mathbb{R}^d$ first concatenates the two inputs along the feature dimension and then apply a 2-layer MLP with SiLU activation.

We prove that both the invariant and equivariant message blocks can be viewed as special instances by extending the invariant self-attention module and the equivariant cross-attention module of our framework respectively. In particular, we extend $\psi_V^I, \psi_V^{E-I}$ introduced in the Section 4.1 to be query-dependent, i.e., $\psi_V^{I,i}, \psi_V^{E\_I,i}$ that depends on the atom $i$'s representations. Concretely, in the invariant self-attention module, we set $\psi_V^{I,i}(\mathbf{z}_j^I) = \phi_s(\mathbf{z}_j^I) \odot \mathcal{W}_s(||\mathbf{r}_i - \mathbf{r}_j||)$. Similarly, in the equivariant cross-attention module, we set $\psi_V^{E\_I,i}(\mathbf{z}_j^I, \mathbf{z}_j^E) = \mathbf{z}_j^E \odot \phi_{vv}(\mathbf{z}_j^I) \cdot \mathcal{W}_{vv}(||\mathbf{r}_i - \mathbf{r}_j||) + \phi_{vs}(\mathbf{z}_j^I) \cdot \mathcal{W}'_{vs} \frac{\mathbf{r}_i - \mathbf{r}_j}{||\mathbf{r}_i - \mathbf{r}_j||}$. In such way, the invariant self-attention module and the equivariant cross-attention module can express the invariant and equivariant message blocks respectively, e.g., the parameters to transform Query and Key are trained/initialized to zero, and the number of atoms can be equipped by initialization, which is necessary to express the sum operator by using the attention as shown in (Ying et al., 2021a).

Moreover, we prove that the update blocks can also be viewed as special instances by extending the FFN blocks in our framework. In particular, we set $\text{Inv-FFN}(\mathbf{z}_i^I) = \mathbf{a}_{ss}(\mathbf{z}_i^I, ||\mathbf{z}_i^E \mathbf{V}||) +$

$\mathbf{a}_{sv}(\mathbf{z}_i^I, ||\mathbf{z}_i^E \mathbf{V}||) < \mathbf{z}_i^E \mathbf{U}, \mathbf{z}_i^E \mathbf{V} >$ and $\text{Equ-FFN}(\mathbf{z}_i^E) = \mathbf{a}_{vv}(\mathbf{z}_i^I, ||\mathbf{z}_i^E \mathbf{V}||) (\mathbf{z}_i^E \mathbf{U})$, then both Inv-FFN and Equ-FFN can express the update blocks. Note that the parameters of the remaining blocks (Inv-Cross-Attn, Equ-Self-Attn) can be trained/initialized to be zero. In such way, the PaiNN model can be instantiated through our design philosophy introduced in Section 4.1.

**TorchMD-Net (Thölke & De Fabritiis, 2022).** Similarly to PaiNN, both invariant representations $\mathbf{Z}^I = [\mathbf{z}_1^{I\top}; ...; \mathbf{z}_n^{I\top}] \in \mathbb{R}^{n \times d}$ and equivariant representations $\mathbf{Z}^E = [\mathbf{z}_1^E; ...; \mathbf{z}_n^E] \in \mathbb{R}^{n \times 3 \times d}$ are maintained in TorchMD-Net, where $\mathbf{z}_i^I \in \mathbb{R}^d$ and $\mathbf{z}_i^E \in \mathbb{R}^{3 \times d}$ are the invariant and equivariant representations for atom $i$, respectively. In each layer, the representations are updated as follows:

$$
\begin{aligned}
\mathbf{Z}'^{I,l} &= \mathbf{Z}^{I,l} + \text{TorchMD-Inv-Block-1}(\mathbf{Z}^{I,l}) \\
\mathbf{Z}^{I,l+1} &= \mathbf{Z}'^{I,l} + \text{TorchMD-Inv-Block-2}(\mathbf{Z}'^{I,l}, \mathbf{Z}^{E,l}) \\
\mathbf{Z}^{E,l+1} &= \mathbf{Z}^{E,l} + \text{TorchMD-Equ-Block}(\mathbf{Z}^{I,l}, \mathbf{Z}^{E,l})
\end{aligned}
\tag{12}
$$

The invariant representations in TorchMD-Inv-Block-1 and TorchMD-Inv-Block-2 are updated as follows. For brevity, we omit the layer index $l$.

$$
\mathbf{Q}_i = W^Q \mathbf{z}_i^I, \mathbf{K}_j = W^K \mathbf{z}_j^I, \mathbf{V}_j^{(1)} = W^{V(1)} \mathbf{z}_j^I
$$

$$
\alpha_{ij} = \text{SiLU}\left(\mathbf{Q}_i^\top (\mathbf{K}_j \circ \mathbf{D}_{ij}^K)\right)
$$

$$
\text{TorchMD-Inv-Block-1}(\mathbf{z}_i^I) = O_1\left(\sum_j \alpha_{ij} \cdot \mathbf{V}_j^{(1)} \circ \mathbf{D}_{ij}^{V(1)}\right)
\tag{13}
$$

$$
\text{TorchMD-Inv-Block-2}(\mathbf{z}_i^I, \mathbf{z}_i^E) = O_2\left(\sum_j \alpha_{ij} \cdot \mathbf{V}_j^{(1)} \circ \mathbf{D}_{ij}^{V(1)}\right) \circ < \mathbf{z}_i^E \mathbf{U}_1, \mathbf{z}_i^E \mathbf{U}_2 >
$$

$\mathbf{W}^Q, \mathbf{W}^K, \mathbf{W}^{V(1)}, \mathbf{U}_1, \mathbf{U}_2 \in \mathbb{R}^{d \times d}$ are learnable parameters. $\circ$ denotes the element-wise product. $\mathbf{D}_{ij}^K, \mathbf{D}_{ij}^{V(1)} : \mathbb{R} \to \mathbb{R}^d$ takes $||\mathbf{r}_i - \mathbf{r}_j||$ as input and uses radial basis functions followed by a non-linear activation to transform it. $O_1, O_2 : \mathbb{R}^d \to \mathbb{R}^d$ are learnable linear transformations. $< \cdot, \cdot >$ is defined the same way as in Section 4.2, i.e., given $x, y \in \mathbb{R}^{3 \times d}, z = < x, y > \in \mathbb{R}^d$, where $z_{[k]} = x_{[:,k]}^\top y_{[:,k]}$. On the other hand, the equivariant representations are updated as follows:

$$
\mathbf{V}_j^{(2)} = \mathbf{W}^{V(2)} \mathbf{z}_j^I, \mathbf{V}_j^{(3)} = \mathbf{W}^{V(3)} \mathbf{z}_j^I
$$

$$
\text{TorchMD-Equ-Block}(\mathbf{z}_i^I, \mathbf{z}_i^E) = \sum_j \left((\mathbf{V}_j^{(2)} \circ \mathbf{D}_{ij}^{V(2)}) \odot \mathbf{z}_j^E + \frac{\mathbf{r}_i - \mathbf{r}_j}{||\mathbf{r}_i - \mathbf{r}_j||}(\mathbf{V}_j^{(3)} \circ \mathbf{D}_{ij}^{V(3)})^\top\right)
\tag{14}
$$

$$
+ O_3\left(\sum_j \alpha_{ij} \cdot \mathbf{V}_j^{(1)} \circ \mathbf{D}_{ij}^{V(1)}\right) \odot \mathbf{z}_i^E \mathbf{U}_3
$$

$\mathbf{W}^{V(2)}, \mathbf{W}^{V(3)}, \mathbf{U}_3 \in \mathbb{R}^{d \times d}$ are learnable parameters. $\circ$ denotes the element-wise product. $\odot$ is defined the same way as in Section 4.2, i.e., given $x \in \mathbb{R}^{3 \times d}, y \in \mathbb{R}^d, z = x \odot y \in \mathbb{R}^{3 \times d}$, where $z_{[i,j]} = x_{[i,k]} \cdot y_{[k]}$. $\mathbf{D}_{ij}^{V(2)}, \mathbf{D}_{ij}^{V(3)} : \mathbb{R} \to \mathbb{R}^d$ takes $||\mathbf{r}_i - \mathbf{r}_j||$ as input and use radial basis functions followed by a non-linear activation to transform it. $O_3 : \mathbb{R}^d \to \mathbb{R}^d$ is a learnable linear transformation. $\frac{\mathbf{r}_i - \mathbf{r}_j}{||\mathbf{r}_i - \mathbf{r}_j||} \in \mathbb{R}^3$ denotes the relative direction between atom $i$'s and $j$'s positions.

We prove that the TorchMD-Inv-Block-1 and TorchMD-Inv-Block-2 can be viewed as special instances by extending the invariant self-attention module and invariant cross-attention module of our framework respectively. Concretely, in the invariant self-attention module, we set $\psi_Q^I(\mathbf{z}_i^I) = W^Q \mathbf{z}_i^I, \psi_K^{I,\,i}(\mathbf{z}_j^I) = W^K \mathbf{z}_j^I \circ \mathbf{D}_{ij}^K, \psi_V^{I,\,i}(\mathbf{z}_j^I) = O_1\left(W^{V(1)} \mathbf{z}_j^I \circ \mathbf{D}_{ij}^{V(1)}\right)$ and use SiLU instead of Softmax for calculating attention probability. By rewriting TorchMD-Inv-Block-1 in the equivalent form $\text{TorchMD-Inv-Block-1}(\mathbf{z}_i^I) = \sum_j \alpha_{ij} \cdot O_1\left(\mathbf{V}_j^{(1)} \circ \mathbf{D}_{ij}^{V(1)}\right)$, the invariant self-attention module can express it by equipping the number of atoms for expressing the sum operation using the attention.

In the invariant cross-attention module, we set $\psi_Q^I(\mathbf{z}_i^I) = W^Q \mathbf{z}_i^I, \psi_K^{I-E,\,i}(\mathbf{z}_j^I, \mathbf{z}_j^E) = W^K \mathbf{z}_j^I \circ \mathbf{D}_{ij}^K, \psi_V^{I-E,\,i}(\mathbf{z}_j^I, \mathbf{z}_j^E) = O_2\left(W^{V(1)} \mathbf{z}_j^I \circ \mathbf{D}_{ij}^{V(1)}\right) \circ < \mathbf{U}_1 \mathbf{z}_i^E, \mathbf{U}_2 \mathbf{z}_i^E >$, and use SiLU instead of

Softmax for calculating attention probability. By rewriting TorchMD-Inv-Block-2 in the equivalent form TorchMD-Inv-Block-2$\left(\mathbf{z}_i^I, \mathbf{z}_i^E\right) = \sum_j \alpha_{ij} \cdot O_2 \left(\mathbf{V}_j^{(1)} \circ \mathbf{D}_{ij}^{V^{(1)}}\right) \circ < U_1 \mathbf{z}_i^E, U_2 \mathbf{z}_i^E >$, the invariant cross-attention module can express it by equipping the number of atoms.

Moreover, we prove that the TorchMD-Equ-Block can be viewed as a special instance by extending the equivariant cross-attention module of our framework. In particular, we set $\psi_V^{E\_I, i}(\mathbf{z}_j^I, \mathbf{z}_j^E) = (W^{V^{(2)}} \mathbf{z}_j^I \circ \mathbf{D}_{ij}^{V^{(2)}}) \odot \mathbf{z}_j^E + \frac{\mathbf{r}_i - \mathbf{r}_j}{||\mathbf{r}_i - \mathbf{r}_j||}(W^{V^{(3)}} \mathbf{z}_j^I \circ \mathbf{D}_{ij}^{V^{(3)}})^\top + \alpha_{ij} \cdot O_3 \left(W^{V^{(1)}} \mathbf{z}_j^I \circ \mathbf{D}_{ij}^{V^{(1)}}\right) \odot U_3 \mathbf{z}_i^E$. By rewriting TorchMD-Equ-Block in the equivalent form TorchMD-Equ-Block$(\mathbf{z}_i^I, \mathbf{z}_i^E) = \sum_j \left((\mathbf{V}_j^{(2)} \circ \mathbf{D}_{ij}^{V^{(2)}}) \odot \mathbf{z}_j^E + \frac{\mathbf{r}_i - \mathbf{r}_j}{||\mathbf{r}_i - \mathbf{r}_j||}(\mathbf{V}_j^{(3)} \circ \mathbf{D}_{ij}^{V^{(3)}})^\top\right) + \sum_j \alpha_{ij} \cdot O_3 \left(\mathbf{V}_j^{(1)} \circ \mathbf{D}_{ij}^{V^{(1)}}\right) \odot U_3 \mathbf{z}_i^E = \sum_j \left((\mathbf{V}_j^{(2)} \circ \mathbf{D}_{ij}^{V^{(2)}}) \odot \mathbf{z}_j^E + \frac{\mathbf{r}_i - \mathbf{r}_j}{||\mathbf{r}_i - \mathbf{r}_j||}(\mathbf{V}_j^{(3)} \circ \mathbf{D}_{ij}^{V^{(3)}})^\top + \alpha_{ij} \cdot O_3 \left(\mathbf{V}_j^{(1)} \circ \mathbf{D}_{ij}^{V^{(1)}}\right) \odot U_3 \mathbf{z}_i^E\right)$, it is straightforward that the equivariant cross-attention module can express the TorchMD-Equ-Block, e.g., the parameters to transform Query and Key are trained/initialized to zero, and the number of atoms can be equipped by initialization. Note that the parameters of the remaining blocks (Equ-Self-Attn, Inv-FFN, Equ-FFN) can be trained/initialized to be zero. In such ways, the TorchMD-Net model can be instantiated through our design philosophy introduced in Section 4.1.

## C.2 EXTENSION TO OTHER GEOMETRIC CONSTRAINTS

In this subsection, we showcase how to extend our framework to encode other geometric constraints. In particular, we consider the $E(3)$ group, which comprises translation, rotation and reflection. Formally, let $V_\mathcal{M}$ denote the space of molecular systems, for each atom $i$, we define equivariant representation $\phi^E$ and invariant representation $\phi^I$ if $\forall\, g = (\mathbf{t}, \mathbf{R}) \in E(3), \mathcal{M} = (\mathbf{X}, R) \in V_\mathcal{M}$, the following conditions are satisfied:

$$\phi^E : V_\mathcal{M} \to \mathbb{R}^{3 \times d}, \qquad \mathbf{R}\phi^E(\mathbf{X}, \{\mathbf{r}_1, ..., \mathbf{r}_n\}) + \mathbf{t}\mathbf{1}^\top = \phi^E(\mathbf{X}, \{\mathbf{R}\mathbf{r}_1 + \mathbf{t}, ..., \mathbf{R}\mathbf{r}_n + \mathbf{t}\}) \quad (15a)$$

$$\phi^I : V_\mathcal{M} \to \mathbb{R}^d, \qquad \phi^I(\mathbf{X}, \{\mathbf{r}_1, ..., \mathbf{r}_n\}) = \phi^I(\mathbf{X}, \{\mathbf{R}\mathbf{r}_1 + \mathbf{t}, ..., \mathbf{R}\mathbf{r}_n + \mathbf{t}\}) \quad (15b)$$

where $\mathbf{t} \in \mathbb{R}^3$ is a translation vector, $\mathbf{R} \in \mathbb{R}^{3 \times 3}, \det(\mathbf{R}) = \pm 1$ is an orthogonal transformation matrix and $\mathbf{X} \in \mathbb{R}^{n \times d}$ denotes the atoms with features, $R = \{\mathbf{r}_1, ..., \mathbf{r}_n\}, \mathbf{r}_i \in \mathbb{R}^3$ denotes the cartesian coordinate of atom $i$. In particular, the additional requirement is to encode the translation and reflection equivariance of the equivariant representations, which can be achieved by modifying the conditions of our framework (Eqn.(4)).

With the invariant representation $\mathbf{Z}^I$ and the equivariant representation $\mathbf{Z}^E$ that satisfy the constraints (Eqn.(15a) and Eqn.(15b)), we separately redefine the conditions of each component. It is worth noting that the reflection invariance is directly satisfied ($\mathbf{R}\mathbf{R}^\top = \mathbf{R}^\top \mathbf{R} = \mathbf{I}$) from the analysis in Section B.1 and Section B.2, which is required in (1) the calculation of attention probability in Equ-Self-Attn, Equ-Cross-Attn; (2) the calculation of $\mathbf{K}^{I\_E}$ and $\mathbf{V}^{I\_E}$. Thus, we only need to encode the translation equivariance constraint. Given the update rules (Eqn.(4)), it can be achieved by simply setting each component (Inv-Self-Attn, Inv-Cross-Attn, Equ-Self-Attn, Equ-Cross-Attn, Inv-FFN, Equ-FFN) to be translation-invariant. In this way, the output equivariant representation can preserve the equivariance to the $E(3)$ group. We extend our framework to achieve this goal, which is introduced below:

**Self-Attention modules.** For Inv-Self-Attn, the condition remains unchanged. For Equ-Self-Attn, the additional condition is that $\psi^E$ should keep the translation invariance. Here we give a simple instantiation: $\mathbf{Q}^E = \psi_Q^E(\mathbf{Z}^E) = (\mathbf{Z}^E - \mu_{\mathbf{Z}^E})W_Q^E, \mathbf{K}^E = \psi_K^E(\mathbf{Z}^E) = (\mathbf{Z}^E - \mu_{\mathbf{Z}^E})W_K^E, \mathbf{V}^E = \psi_V^E(\mathbf{Z}^E) = (\mathbf{Z}^E - \mu_{\mathbf{Z}^E})W_V^E$, where $\mu_{\mathbf{Z}^E, i} = \frac{1}{d}\sum_{k=1}^n \mathbf{Z}^E_{[i,:,k]}\mathbf{1}^\top$.

**Cross-Attention modules.** For Inv-Cross-Attn, the condition for $\psi^I$ remains unchanged, while $\psi^{I\_E}$ should keep the translation invariance. For Equ-Cross-Attn, both $\psi^E$ and $\psi^{E\_I}$ are required to be translation-invariant. Here we give an instantiation: $\mathbf{Q}^E = \psi_Q^E(\mathbf{Z}^E) = (\mathbf{Z}^E - \mu_{\mathbf{Z}^E})W_Q^E$, and

$$\begin{aligned}
\mathbf{K}^{I\_E} &=< (\mathbf{Z}^E - \mu_{\mathbf{Z}^E})W_{K,1}^{I\_E}, (\mathbf{Z}^E - \mu_{\mathbf{Z}^E})W_{K,2}^{I\_E} >, \quad \mathbf{V}^{I\_E} =< (\mathbf{Z}^E - \mu_{\mathbf{Z}^E})W_{V,1}^{I\_E}, (\mathbf{Z}^E - \mu_{\mathbf{Z}^E})W_{V,2}^{I\_E} > \\
\mathbf{K}^{E\_I} &= (\mathbf{Z}^E - \mu_{\mathbf{Z}^E})W_{K,1}^{E\_I} \odot \mathbf{Z}^I W_{K,2}^{E\_I}, \qquad \mathbf{V}^{E\_I} = (\mathbf{Z}^E - \mu_{\mathbf{Z}^E})W_{V,1}^{E\_I} \odot \mathbf{Z}^I W_{V,2}^{E\_I}
\end{aligned}$$

$$(16)$$

**Feed-Forward Networks.** Similarly, the condition for Inv-FFN remains unchanged. For Equ-FFN, it also should keep the translation invariance, e.g., Equ-FFN($\mathbf{Z}''^E$) = $((\mathbf{Z}''^E - \mu_{\mathbf{Z}^E})W_1^E \odot \text{GELU}(\mathbf{Z}''^I W_2^I))W_3^E$.

**Remark.** With the above additional conditions, our framework can additionally be extended to encode geometric constraints towards $E(3)$ group. Note that the design of the input layer should also encode the constraints (Eqn.(15a) and Eqn.(15b)). For example, the invariant representation remains unchanged as $\mathbf{Z}^{I,0} = \mathbf{X}$. while the equivariant representation can be directly set as $\mathbf{Z}_i^{E,0} = \mathbf{r}_i$. In this way, the geometric constraints are well satisfied.

# D EXPERIMENTAL DETAILS

## D.1 OC20 IS2RE

**Baselines.** We compare our GeoMFormer with several competitive baselines for learning geometric molecular representations. Crystal Graph Convolutional Neural Network (CGCNN) (Xie & Grossman, 2018) developed novel approaches to modeling periodic crystal systems with diverse features as node embeddings. SchNet (Schütt et al., 2018) leveraged the interatomic distances encoded via radial basis functions, which serve as the weights of continuous-filter convolutional layers. DimeNet++ (Gasteiger et al., 2020a) introduced the directional message passing that encodes both distance and angular information between triplets of atoms.

GemNet (Gasteiger et al., 2021) embedded all atom pairs within a given cutoff distance based on inter-atomic directions, and proposed three forms of interaction to update the directional embeddings: Two-hop geometric message passing (Q-MP), one-hop geometric message passing (T-MP), and atom self-interactions. An efficient variant named GemNet-T is proposed to use cheaper forms of interaction.

SphereNet (Liu et al., 2022b) used the spherical coordinate system to represent the relative location of each atom in the 3D space and proposed the spherical message passing. GNS (Pfaff et al., 2020) is a framework for learning mesh-based simulations using graph neural networks and can handle complex physical systems. Graphormer-3D (Shi et al., 2022) extended Graphormer(Ying et al., 2021a) to learn geometric molecular representations, which encodes the interatomic distance as attention bias terms and performed well on large-scale datasets. Equiformer (Liao & Smidt, 2022) uses the tensor product operations to build a new scalable equivariant Transformer architecture and outperforms strong baselines on the large-scale OC20 dataset (Chanussot et al., 2021).

**Settings.** As introduced in Section 5.1.1, we follow the experimental setup of Graphormer-3D (Shi et al., 2022) for a fair comparison. Our GeoMFormer model consists of 12 layers. The dimension of hidden layers and feed-forward layers is set to 768. The number of attention heads is set to 48. The number of Gaussian Basis kernels is set to 128. We use AdamW as the optimizer and set the hyper-parameter $\epsilon$ to 1e-6 and $(\beta_1, \beta_2)$ to (0.9,0.98). The gradient clip norm is set to 5.0. The peak learning rate is set to 2e-4. The batch size is set to 128. The dropout ratios for the input embeddings, attention matrices, and hidden representations are set to 0.0, 0.1, and 0.0 respectively. The weight decay is set to 0.0. The model is trained for 1 million steps with a 60k-step warm-up stage. After the warm-up stage, the learning rate decays linearly to zero. Following Liao & Smidt (2022), we also use the noisy node data augmentation strategy (Godwin et al., 2021) to improve the performance. The model is trained on 16 NVIDIA Tesla V100 GPUs.

## D.2 OC20 IS2RS

**Baselines.** In this experiment, we choose several competitive baselines that perform well on equivariant prediction tasks for molecules. PaiNN (Schütt et al., 2021) built upon the framework of EGNN (Satorras et al., 2021) to maintain both invariant and equivariant representations and further used the Hardamard product operation to transform the equivariant representations. Specialized tensor prediction blocks were also developed for different molecular properties. TorchMD-Net (Thölke & De Fabritiis, 2022) developed an equivariant Transformer architecture by using similar Hardamard product operations and achieved strong performance on various tasks.

SpinConv (Shuaibi et al., 2021) encoded angular information with a local reference frame defined by two atoms and used a spin convolution on the spherical representation to capture rich angular information while maintaining rotation invariance. An additional prediction head is used to perform the equivariant prediction task, GemNet-dT (Gasteiger et al., 2021) is a variant of GemNet-T that can directly perform force prediction and other equivariant tasks, e.g., the relaxed positions in this experiment. GemNet-OC (Gasteiger et al., 2022) is an extension of GemNet by using more efficient components and achieved better performance on OC20 tasks.

**Settings.** As introduced in Section 5.1.2, we adopt the direct prediction setting for comparing the ability to perform equivariant prediction tasks on OC20 IS2RS. In particular, we re-implemented the baselines and carefully trained these models for a fair comparison. Our GeoMFormer model consists of 12 layers. The dimension of hidden layers and feed-forward layers is set to 768. The number of attention heads is set to 48. The number of Gaussian Basis kernels is set to 128. We use AdamW as the optimizer and set the hyper-parameter $\epsilon$ to 1e-6 and $(\beta_1, \beta_2)$ to (0.9,0.98). The gradient clip norm is set to 5.0. The peak learning rate is set to 2e-4. The batch size is set to 64. The dropout ratios for the input embeddings, attention matrices, and hidden representations are set to 0.0, 0.1, and 0.0 respectively. The weight decay is set to 0.0. The model is trained for 1 million steps with a 60k-step warm-up stage. After the warm-up stage, the learning rate decays linearly to zero. The model is trained on 16 NVIDIA Tesla V100 GPUs.

## D.3  PCQM4Mv2

**Baselines.** We compare our GeoMFormer with several competitive baselines from the leaderboard of OGB Large-Scale Challenge (Hu et al., 2021). First, we compare several message-passing neural network (MPNN) variants. Two widely used models, GCN (Kipf & Welling, 2016) and GIN (Xu et al., 2019) are compared along with their variants with virtual node (VN) (Gilmer et al., 2017; Hu et al., 2020). Besides, we compare GINE-VN (Brossard et al., 2020) and DeeperGCN-VN (Li et al., 2020). GINE is the multi-hop version of GIN. DeeperGCN is a 12-layer GNN model with carefully designed aggregators. The result of MLP-Fingerprint (Hu et al., 2021) is also reported. The complexity of these models is generally $\mathcal{O}(n)$, where $n$ denotes the number of atoms.

Additionally, we compare with several Graph Transformer models, whose computational complexity is $\mathcal{O}(n^2)$. TokenGT (Kim et al., 2022) purely used node and edge representations as the input and adopted the standard Transformer architecture without graph-specific modifications. EGT (Hussain et al., 2022) used global self-attention as an aggregation mechanism and utilized edge channels to capture structural information. GRPE (Park et al., 2022) considered both node-spatial and node-edge relations and proposed a graph-specific relative positional encoding. Graphormer (Ying et al., 2021a) developed graph structural encodings and integrated them into a standard Transformer model, which achieved impressive performance across several world competitions (Ying et al., 2021b; Shi et al., 2022). GraphGPS (Rampavsek et al., 2022) proposed a framework to integrate the positional and structural encodings, local message-passing mechanism, and global attention mechanism into the Transformer model. All these models are designed to learn 2D molecular representations.

There also exist several models capable of utilizing the 3D geometric structure information in the training set of PCQM4Mv2. Transformer-M (Luo et al., 2022) is a Transformer-based Molecular model that can take molecular data of 2D or 3D formats as input and learn molecular representations, which was widely adopted by the winners of the 2nd OGB Large-Scale Challenge. GPS++ (Masters et al., 2022) is a hybrid MPNN and Transformer model built on the GraphGPS framework (Rampavsek et al., 2022). It follows Transformer-M to utilize 3D atom positions and auxiliary tasks to win first place in the large-scale challenge.

Last, we include two complex models with $\mathcal{O}(n^3)$ complexity. GEM-2 (Liu et al., 2022a) used multiple branches to encode the full-range interactions between many-body objects and designed an axial attention mechanism to efficiently approximate the interaction with low computational cost. Uni-Mol+ (Lu et al., 2023) proposed an iterative prediction framework to achieve accurate quantum property prediction. It first generated 3D geometric structures from the 2D molecular graph using fast yet inaccurate methods, e.g., RDKit (Landrum, 2016). Given the inaccurate 3D structure as the input, the model is required to predict the equilibrium structure in an iterative manner. The predicted equilibrium structure is used to predict the quantum property. Uni-Mol+ simultaneously maintain both atom representations and pair representations, which induce the triplet complexity

when updating the pair representations. With the carefully designed training strategy, Uni-Mol+ achieves state-of-the-art performance on PCQM4Mv2 while yielding high computational costs.

**Settings.** As previously stated, DFT-calculated equilibrium geometric structures are provided for molecules in the training set. The molecules in the validation set do not have such information. We follow Uni-Mol+ (Lu et al., 2023) to train our GeoMFormer. In particular, our model takes the RDKit-generated geometric structures as the input and is required to predict both the HOMO-LUMO energy gap and the equilibrium structure by leveraging invariant and equivariant representations respectively. After training, the model is able to predict the HOMO-LUMO gap using the RDKit-generated geometric structures. We refer the readers to Uni-Mol+ (Lu et al., 2023) for more details on the training strategies.

Our GeoMFormer model consists of 8 layers. The dimension of hidden layers and feed-forward layers is set to 512. The number of attention heads is set to 32. The number of Gaussian Basis kernels is set to 128. We use AdamW as the optimizer, and set the hyper-parameter $\epsilon$ to 1e-8 and $(\beta_1, \beta_2)$ to (0.9,0.999). The gradient clip norm is set to 5.0. The peak learning rate is set to 2e-4. The batch size is set to 1024. The dropout ratios for the input embeddings, attention matrices, and hidden representations are set to 0.0, 0.1, and 0.1 respectively. The weight decay is set to 0.0. The model is trained for 1.5 million steps with a 150k-step warm-up stage. After the warm-up stage, the learning rate decays linearly to zero. Other hyper-parameters are kept the same as the Uni-Mol+ for a fair comparison. The model is trained on 16 NVIDIA Tesla V100 GPUs.

### D.4  MOLECULE3D

**Baselines.**  We follow (Wang et al., 2022) to use several competitive baselines for comparison including GIN-Virtual (Hu et al., 2021), SchNet (Schütt et al., 2018), DimeNet++ (Gasteiger et al., 2020a), SphereNet (Liu et al., 2022b) which have already been introduced in previous sections. ComENet (Wang et al., 2022) proposed a message-passing layer that operates within the 1-hop neighborhood of atoms and encoded the rotation angles to fulfill global completeness. We also implement both PaiNN (Schütt et al., 2021) and TorchMD-Net (Thölke & De Fabritiis, 2022) for comparisons.

**Settings.**  Following (Wang et al., 2022), we evaluate our GeoMFormer model on both random and scaffold splits. Our GeoMFormer model consists of 12 layers. The dimension of hidden layers and feed-forward layers is set to 768. The number of attention heads is set to 48. The number of Gaussian Basis kernels is set to 128. We use AdamW as the optimizer, and set the hyper-parameter $\epsilon$ to 1e-8 and $(\beta_1, \beta_2)$ to (0.9,0.999). The gradient clip norm is set to 5.0. The peak learning rate is set to 3e-4. The batch size is set to 1024. The dropout ratios for the input embeddings, attention matrices, and hidden representations are set to 0.0, 0.1, and 0.1 respectively. The weight decay is set to 0.0. The model is trained for 1 million steps with a 60k-step warm-up stage. After the warm-up stage, the learning rate decays linearly to zero. The model is trained on 16 NVIDIA V100 GPUs.

### D.5  N-BODY SIMULATION

**Baselines.**  Following (Satorras et al., 2021), we choose several competitive baselines for comparison. Radial Field (Köhler et al., 2019) developed theoretical tools for constructing equivariant flows and can be used to perform equivariant prediction tasks. Tensor Field Network (Thomas et al., 2018) embedded the position of an object in the Cartesian space into higher-order representations via products between learnable radial functions and spherical harmonics. In SE(3)-Transformer (Fuchs et al., 2020), the standard attention mechanism was adapted to equivariant features using operations in the Tensor Field Network model. EGNN (Satorras et al., 2021) proposed a simple framework. Its invariant representations encode type information and relative distance, and are further used in vector scaling functions to transform the equivariant representations.

**Settings.**  The input of the model includes initial positions $\mathbf{p}^0 = \{\mathbf{p}_1^0, \ldots, \mathbf{p}_5^0\} \in \mathbb{R}^{5 \times 3}$ of five objects, and their initial velocities $\mathbf{v}^0 = \{\mathbf{v}_1^0, \ldots, \mathbf{v}_5^0\} \in \mathbb{R}^{5 \times 3}$ and respective charges $\mathbf{c} = \{c_1, \ldots, c_5\} \in \{-1, 1\}^5$. We encode positions and velocities via separate equivariant streams, and updated them with separate invariant representations via cross-attention modules. The equivariant prediction is based on both equivariant representations.

Table 7: Results on MD trajectories from the MD17 dataset. Scores are given by the MAE of energy predictions (kcal/mol) and forces (kcal/mol/Å). NequIP does not provide errors on energy, for PaiNN we include the results with lower force error out of training only on forces versus on forces and energy. Benzene corresponds to the dataset originally released in Chmiela et al. (2017), which is sometimes left out from the literature. Our results are averaged over three random splits.

| Molecule | | SchNet | PhysNet | DimeNet | PaiNN | NequIP | TorchMD-Net | GeoMFormer |
|---|---|---|---|---|---|---|---|---|
| Aspirin | *energy* | 0.37 | 0.230 | 0.204 | 0.167 | - | 0.123 | **0.118** |
| | *forces* | 1.35 | 0.605 | 0.499 | 0.338 | 0.348 | 0.253 | **0.171** |
| Benzene | *energy* | 0.08 | - | 0.078 | - | - | 0.058 | **0.052** |
| | *forces* | 0.31 | - | 0.187 | - | 0.187 | 0.196 | **0.146** |
| Ethanol | *energy* | 0.08 | 0.059 | 0.064 | 0.064 | - | 0.052 | **0.047** |
| | *forces* | 0.39 | 0.160 | 0.230 | 0.224 | 0.208 | 0.109 | **0.062** |
| Malondialdehyde | *energy* | 0.13 | 0.094 | 0.104 | 0.091 | - | 0.077 | **0.071** |
| | *forces* | 0.66 | 0.319 | 0.383 | 0.319 | 0.337 | 0.169 | **0.133** |
| Naphthalene | *energy* | 0.16 | 0.142 | 0.122 | 0.116 | - | 0.085 | **0.081** |
| | *forces* | 0.58 | 0.310 | 0.215 | 0.077 | 0.097 | 0.061 | **0.040** |
| Salicylic Acid | *energy* | 0.20 | 0.126 | 0.134 | 0.116 | - | **0.093** | 0.099 |
| | *forces* | 0.85 | 0.337 | 0.374 | 0.195 | 0.238 | 0.129 | **0.098** |
| Toluene | *energy* | 0.12 | 0.100 | 0.102 | 0.095 | - | **0.074** | 0.078 |
| | *forces* | 0.57 | 0.191 | 0.216 | 0.094 | 0.101 | 0.067 | **0.041** |
| Uracil | *energy* | 0.14 | 0.108 | 0.115 | 0.106 | - | **0.095** | **0.095** |
| | *forces* | 0.56 | 0.218 | 0.301 | 0.139 | 0.173 | 0.095 | **0.068** |

We follow the settings in (Satorras et al., 2021) for a fair comparison. Our GeoMFormer model consists of 4 layers. The dimension of hidden layers and feed-forward layers is set to 80. The number of attention heads is set to 8. The number of Gaussian Basis kernels is set to 64. We use Adam as the optimizer, and set the hyper-parameter $\epsilon$ to 1e-8 and $(\beta_1, \beta_2)$ to (0.9,0.999). The learning rate is fixed to 3e-4. The batch size is set to 100. The dropout ratios for the input embeddings, attention matrices, activation functions, and hidden representations are all set to 0.4, and the drop path probability is set to 0.4. The model is trained for 10,000 epochs. The number of training samples is set to 3.000. The model is trained on 1 NVIDIA V100 GPUs.

## D.6 MD17

MD17 (Xu et al., 2021) consists of molecular dynamics trajectories of several small organic molecules. Each molecule has its geometric structure along with the corresponding energy and force. The task is to predict both the energy and force of the molecule's geometric structure in the current state. To evaluate the performance of models in a limited data setting, all models are trained on only 1,000 samples from which 50 are used for validation. The remaining data is used for evaluation. For each molecule, we train a separate model on data samples of this molecule only. We set the model parameter budget the same as Thölke & De Fabritiis (2022). Following (Thölke & De Fabritiis, 2022), we compare our GeoMFormer with several competitive baselines: (1) SchNet (Schütt et al., 2018); (2) PhysNet (Unke & Meuwly, 2019); (3) DimeNet (Gasteiger et al., 2020b); (4) PaiNN (Schütt et al., 2021); (5) NequIP (Batzner et al., 2022); (6) TorchMD-Net (Thölke & De Fabritiis, 2022). The results are presented in Table 7. It can be easily seen that our GeoMFormer achieves competitive performance on the energy prediction task (5 best and 1 tie out of 8 molecules) and consistently outperforms the best baselines by a significantly large margin on the force prediction task, i.e., 30.6% relative force MAE reduction in average.

## D.7 MORE ANALYSIS

In this subsection, we conduct comprehensive experiments for ablation studies on each building component of our GeoMFormer model, including both self-attention and cross-attention modules (Inv-Self-Attn, Equ-Self-Attn, Inv-Cross-Attn, Equ-Cross-Attn), feed-forward networks (Inv-FFN, Equ-FFN), layer normalizations (Inv-LN, Equ-LN) and the structural encoding. Without loss of generality, we conduct the experiments on the N-body Simulation task.

Table 8: Impact of the attention modules on GeoMFormer. All other hyperparameters are kept the same for a fair comparison.

| Inv-Self-Attn | Inv-Cross-Attn | Equ-Self-Attn | Equ-Cross-Attn | MSE ↓ |
|---|---|---|---|---|
| ✓ | ✓ | ✓ | ✓ | 0.0047 |
| ✗ | ✓ | ✓ | ✓ | 0.0051 |
| ✓ | ✗ | ✓ | ✓ | 0.0051 |
| ✓ | ✓ | ✗ | ✓ | 0.0056 |
| ✓ | ✓ | ✓ | ✗ | 0.0054 |
| ✗ | ✓ | ✓ | ✗ | 0.0054 |
| ✓ | ✗ | ✓ | ✗ | 0.0057 |
| ✗ | ✓ | ✗ | ✓ | 0.0055 |
| ✓ | ✗ | ✗ | ✓ | 0.0057 |
| ✗ | ✗ | ✓ | ✗ | 0.0059 |

Table 9: Impact of the FFN modules on GeoMFormer. All other hyperparameters are kept the same for a fair comparison.

Table 10: Impact of the LN modules on GeoMFormer. All other hyperparameters are kept the same for a fair comparison.

Table 11: Impact of structural encoding on GeoMFormer. All other hyperparameters are kept the same for a fair comparison.

| Inv-FFN | Equ-FFN | MSE ↓ |
|---|---|---|
| ✓ | ✓ | 0.0047 |
| ✗ | ✓ | 0.0049 |
| ✓ | ✗ | 0.0055 |
| ✗ | ✗ | 0.0057 |

| Inv-LN | Equ-LN | MSE ↓ |
|---|---|---|
| ✓ | ✓ | 0.0047 |
| ✗ | ✓ | 0.0051 |
| ✓ | ✗ | 0.0077 |
| ✗ | ✗ | 0.0073 |

| Structural Encoding | MSE ↓ |
|---|---|
| ✓ | 0.0047 |
| ✗ | 0.0072 |

**Impact of the attention modules.** As stated in Section 4, our GeoMFormer model consists of four attention modules. We conduct a series of ablation studies to evaluate their contribution to the overall performance. In particular, we consider all possible ablation configurations that involve ablating one or more of the four modules. Note that this is an equivariant prediction task, necessitating the preservation of at least one equivariant attention module. The results are presented in Table 8, which indicates that all four attention modules consistently contribute to boosting the model's performance.

**Impact of the FFN.** We perform ablation studies to ascertain the contribution of both invariant and equivariant FFN modules to the model's performance. Specifically, we examine all possible settings involving the ablation of one or both of the FFN modules. The results are presented in Table 9, which demonstrates that both FFN modules positively contribute to enhancing performance.

**Impact of the LN.** We employ invariant and equivariant LN to stabilize training. To investigate whether the invariant and equivariant LN modules improve performance, we conduct ablation studies that encompass all possible settings of ablating one or both LN modules. The results are displayed in Table 10, demonstrating that both LN modules help to enhance performance.

**Impact of the Structural Encoding.** We incorporate the structural encoding as a bias term when calculating attention probability in our GeoMFormer, as described in Section A. We conduct ablation studies to see if it helps boost performance. Results are shown in Table 11. It can be seen that the introduction of structural encoding leads to improved performance.

# E  BROADER IMPACTS AND LIMITATIONS

This work newly proposes a general framework to learn geometric molecular representations, which has great significance in molecular modeling. Our model has demonstrated considerable positive potential for various physical and chemical applications, such as catalyst discovery and optimization, which can significantly contribute to the advancement of renewable energy processes. However, it is essential to acknowledge the potential negative impacts including the development of toxic drugs and materials. Thus, stringent measures should be implemented to mitigate these risks.

There also exist some limitations to our work. Serving as a general architecture, the ability to scale up both the model and dataset sizes is of considerable interest to the community, which has been partially explored in our extensive experiment. Additionally, our model can also be extended to encompass additional downstream invariant and equivariant tasks, which we have earmarked for future research.

