# OpenReview forum: "GeoMFormer: A General Architecture for Geometric Molecular Representation Learning"
_ICLR.cc/2024/Conference — Submitted to ICLR 2024_

### Official Review · Reviewer_K4hR · 2023-10-29

**Soundness:** 3 good
**Presentation:** 3 good
**Contribution:** 3 good
**Rating:** 6
**Confidence:** 4

**Summary:**

This work presents a Transformer for molecular property prediction. The framework roots from separating invariant/equivariant representations and introduces cross-attention to fuse the two representations effectively. The framework provides a recipe for scaling up the number of parameters easily. Experiments on the multiple benchmarks (OC20, PCQM4Mv2, Molecule3D, MD17, etc) demonstrate SoTA or near SoTA performance.

**Strengths:**

1. The work stems from designing architectures that separate in-/equi-variant representations which provides flexibility of scaling up the # parameters through introducing the cross-attention between the two representations.
2. Proposed GeoMFormer demonstrates SoTA or at lest competitive performance on multiple standard benchmarks when compared with strong baselines.
3. The paper is well written and easy to follow.

**Weaknesses:**

1. The model achieves strong performance in many benchmarks but also contains more # parameters than most baseline models.

**Questions:**

1. Following the weakness, training GeMFormer of different sizes can be useful to elaborate how much performance gain stems from scaling up # parameters. Since the proposed method designs an equivariant Transformer that can be better scaled, such experiments can further demonstrate the advantage of GeMFormer.
2. Is cutoff of interatomic distances applied when implementing attention between atoms in GeMFormer? If yes, how is the cutoff determined?
3. The work has included benchmarks on multiple standard molecular property prediction tasks. How does the model perform on QM9. Though it's a bit older compared with some benchmarks, it's still a widely used. It would be helpful to see where GeMFormer sits on QM9 as well.
4. In appendix D.7, the authors present ablation studies on N-body simulation task. However, as the work focuses mostly on molecular property predictions. It would be better to include some ablations on molecular benchmarks to further validation the design choices.

---

> ### Author Response · Authors · 2023-11-22
> **Response to Reviewer K4hR (Part 1/3)**
>
> Thank you for the recognition of the effectiveness, scalibility and unification achieved by our GeoMFormer. We also appreciate your comments to further improve our work. Here are our responses to your questions.
>
> > **Provide experiments with GeoMFormer of different sizes to see its scalibility (W1 & Q1)**
>
> Thank you for the suggestion. We have carefully followed your advice and conducted preliminary experiments with our GeoMFormer of different sizes on the OC20 IS2RE task. To investigate the scalibility of our GeoMFormer, we vary the number of layers in [6, 12, 18] and keep all other hyperparameters the same for a fair comparison. The results are shown in Table 1.
>
> **Table 1. Performance of GeoMFormer of different sizes on OC20 IS2RE.**
>
>
> | #Layer | ID MAE$\downarrow$ | OOD Ads. MAE$\downarrow$ | **OOD Cat. MAE$\downarrow$** | **OOD Both MAE$\downarrow$** | **Average MAE$\downarrow$** | **ID EwT$\uparrow$** | OOD Ads. EwT$\uparrow$ | OOD Cat. EwT$\uparrow$ | OOD Both EwT$\uparrow$ | **Average EwT$\uparrow$** |
> | :----: | :----------------: | :----------------------: | :--------------------------: | :--------------------------: | :-------------------------: | :------------------: | :--------------------: | :--------------------: | :--------------------: | :-----------------------: |
> |   6    |       0.3986       |          0.4771          |            0.4164            |            0.4266            |           0.4297            |        10.41         |          6.63          |          9.32          |          6.30          |           8.17            |
> |   12   |       0.3883       |          0.4562          |            0.4037            |            0.4083            |           0.4141            |        11.26         |          6.70          |          9.97          |          6.42          |           8.59            |
>
>
>
> From Table 1, we can see that scaling GeoMFormer up brings consistent performance gains on the challenging OC20 IS2RE task: (1) Average MAE: 3.6% relative improvements; (2) Average EwT: 5.1% relative improvements. Moreover, due to the time limits and computational resource constraints, the experiments on our 18-layer GeoMFormer are still running, while we have observed gains comparing its validation curve with the 12-layer GeoMFormer. We will definitely follow your suggestion to conduct further experiments on the effects of the scaling up on other tasks and include these results in our paper as supporting evidence on the scalibility of our GeoMFormer.

---

> ### Author Response · Authors · 2023-11-22
> **Response to Reviewer K4hR (Part 2/3)**
>
> > **Whether apply the cutoff of interatomic distances (Q2)**
>
> In our experiments, we do not apply the cutoff of interatomic distances, which means that each atom can attend to all other atoms. It is a good point when we deal with large molecular systems, and our model can naturally adopt the cutoff of interatomic distances when necessary by (1) set a cutoff of interatomic distances; (2) use this cutoff to determine the neighborhood of each atom; (3) generate attention masks according to the neighborhood of each atom.
>
> > **Regarding experiments on the QM9 dataset (Q3)**
>
> Thank you for your suggestion. We also agree with you on the popularity of the QM9 dataset. This dataset offers quantum chemical properties (e.g., HOMO-LUMO gap, total energy, etc) at Density Functional Theory level for a relevant, consistent, and comprehensive chemical space of 134k small organic molecules with equilibrium structures. Therefore, the tasks of the QM9 dataset are invariant prediction tasks. In our experiments, we have actually adopted several popular datasets that evaluate molecular model in a similar way. For example, the Molecule3D dataset consists of around 3.8 million molecules in total and the task is to predict the HOMO-LUMO energy gap given the equilibrium structures, which is similar to the QM9 task and is much larger in terms of scales. Additionally, the experiments on MD17, which is another fairly popular dataset, also examine the model's ability to perform the energy prediction task. Moreover, we also follow your suggestion to further conduct experiments on the QM9 dataset. Due to the time and computational resources limits, we will update the results on the QM9 dataset to further improve the quality of our work.
>
> > **Provide additional ablation study on molecular benchmarks (Q4)**
>
> Thank you for the suggestion. We have carefully followed your advice and conducted additional ablation study on the MD17 dataset, which is widely used to verify the effectiveness of molecular models. We present our results in Table 2. The results further demonstrate the effectiveness of our design choice. Specifically, removing any of the attention module consistently induces performance drop. In addition, the gains brought by cross-attn modules are extremely significant for both energy prediction (invariant) and force prediction (equivariant): 20.8% relative improvement for energy and 60.8% relative improvement for force gaining form adding the Inv-Cross-Attn and Equ-Cross-Attn, which highlights the effectiveness of the key idea of our design philosophy. We will update this new additional ablation study in the revised version of our paper.

---

> ### Author Response · Authors · 2023-11-22
> **Response to Reviewer K4hR (Part 3/3)**
>
> **Table 2. Ablation Studies on MD17.**
> **Energy**:
>
> | Inv-Self-Attn | Inv-Cross-Attn | Equ-Self-Attn | Equ-Cross-Attn |  Aspirin  |  Benzene  |  Ethanol  | Malondialdehyde | Naphthalene | Salicylic Acid |  Toluene  |  Uracil   |
> | :-----------: | :------------: | :-----------: | :------------: | :-------: | :-------: | :-------: | :-------------: | :---------: | :------------: | :-------: | :-------: |
> | $\checkmark$  |  $\checkmark$  | $\checkmark$  |  $\checkmark$  | **0.118** | **0.052** | **0.047** |    **0.071**    |  **0.081**  |   **0.099**    | **0.078** | **0.095** |
> |   $\times$    |  $\checkmark$  | $\checkmark$  |  $\checkmark$  |   0.156   |   0.063   |   0.058   |      0.085      |    0.092    |     0.115      |   0.090   |   0.102   |
> | $\checkmark$  |    $\times$    | $\checkmark$  |  $\checkmark$  |   0.161   |   0.062   |   0.060   |      0.086      |    0.111    |     0.113      |   0.094   |   0.101   |
> | $\checkmark$  |  $\checkmark$  |   $\times$    |  $\checkmark$  |   0.131   |   0.059   |   0.052   |      0.081      |    0.084    |     0.104      |   0.085   |   0.097   |
> | $\checkmark$  |  $\checkmark$  | $\checkmark$  |    $\times$    |   0.143   |   0.056   |   0.051   |      0.078      |    0.097    |     0.106      |   0.081   |   0.099   |
> | $\checkmark$  |    $\times$    |   $\times$    |  $\checkmark$  |   0.169   |   0.062   |   0.061   |      0.087      |    0.113    |     0.115      |   0.094   |   0.103   |
> | $\checkmark$  |    $\times$    | $\checkmark$  |    $\times$    |   0.172   |   0.064   |   0.061   |      0.086      |    0.112    |     0.116      |   0.095   |   0.103   |
> |   $\times$    |  $\checkmark$  |   $\times$    |  $\checkmark$  |   0.184   |   0.069   |   0.064   |      0.094      |    0.121    |     0.124      |   0.102   |   0.107   |
> |   $\times$    |  $\checkmark$  | $\checkmark$  |    $\times$    |   0.181   |   0.071   |   0.067   |      0.093      |    0.118    |     0.121      |   0.101   |   0.109   |
> | $\checkmark$  |    $\times$    |   $\times$    |    $\times$    |   0.193   |   0.076   |   0.071   |      0.097      |    0.126    |     0.129      |   0.105   |   0.113   |
>
>
>
> **Forces**:
>
> | Inv-Self-Attn | Inv-Cross-Attn | Equ-Self-Attn | Equ-Cross-Attn |  Aspirin  |  Benzene  |  Ethanol  | Malondialdehyde | Naphthalene | Salicylic Acid |  Toluene  |  Uracil   |
> | ------------- | -------------- | ------------- | :------------: | :-------: | :-------: | :-------: | :-------------: | :---------: | :------------: | :-------: | :-------: |
> | $\checkmark$  | $\checkmark$   | $\checkmark$  |  $\checkmark$  | **0.171** | **0.146** | **0.062** |    **0.133**    |  **0.040**  |   **0.098**    | **0.041** | **0.068** |
> | $\times$      | $\checkmark$   | $\checkmark$  |  $\checkmark$  |   0.223   |   0.152   |   0.086   |      0.162      |    0.051    |     0.117      |   0.062   |   0.079   |
> | $\checkmark$  | $\times$       | $\checkmark$  |  $\checkmark$  |   0.257   |   0.159   |   0.104   |      0.178      |    0.063    |     0.126      |   0.076   |   0.091   |
> | $\checkmark$  | $\checkmark$   | $\times$      |  $\checkmark$  |   0.292   |   0.167   |   0.143   |      0.311      |    0.079    |     0.203      |   0.096   |   0.134   |
> | $\checkmark$  | $\checkmark$   | $\checkmark$  |    $\times$    |   0.281   |   0.160   |   0.167   |      0.272      |    0.094    |     0.185      |   0.081   |   0.113   |
> | $\times$      | $\checkmark$   | $\checkmark$  |    $\times$    |   0.313   |   0.164   |   0.196   |      0.319      |    0.134    |     0.191      |   0.105   |   0.195   |
> | $\checkmark$  | $\times$       | $\checkmark$  |    $\times$    |   0.366   |   0.187   |   0.212   |      0.352      |    0.159    |     0.264      |   0.161   |   0.237   |
> | $\times$      | $\checkmark$   | $\times$      |  $\checkmark$  |   0.324   |   0.173   |   0.189   |      0.331      |    0.129    |     0.237      |   0.127   |   0.167   |
> | $\checkmark$  | $\times$       | $\times$      |  $\checkmark$  |   0.358   |   0.182   |   0.219   |      0.337      |    0.172    |     0.272      |   0.134   |   0.241   |
> | $\times$      | $\times$       | $\checkmark$  |    $\times$    |   0.411   |   0.194   |   0.227   |      0.361      |    0.188    |     0.289      |   0.173   |   0.258   |
>
> We thank you again for your efforts in reviewing our paper. We have carefully replied to each of your questions and followed your suggestion to provide additional supporting results on our GeoMFormer. We sincerely look forward to your re-evaluation of our submission based on our clarifications and updated results. May you have any further questions, please tell us and we are willing to address your concerns.

---

> > ### Comment · Reviewer_K4hR · 2023-11-22
> > **Official Comment by Reviewer K4hR**
> >
> > I thank the authors' efforts in answering my questions. I believe the experiment of different model sizes and extra ablation studies further validate the effectiveness of the proposed method. Therefore I stay positive about the work.

---

> > > ### Author Response · Authors · 2023-11-23
> > > **Thank you for the prompt response**
> > >
> > > We sincerely appreciate your prompt response. We also agree with you that the updated experiments of different model sizes and extra ablation studies improve the quality of our work, and we will definitely include them in the revised version of our paper. Thank you again for your efforts in reviewing our paper and the suggestions to improve our work.

---

### Official Review · Reviewer_awZP · 2023-11-01

**Soundness:** 4 excellent
**Presentation:** 4 excellent
**Contribution:** 4 excellent
**Rating:** 8
**Confidence:** 4

**Summary:**

This paper proposes a new Transformer-based architecture, GeoMFormer, for molecular modeling that considers both invariant and equivariant properties in the tasks. Specifically, GeoMFormer comes with two streams: one captures invariance, and the other captures equivariance. A cross-attention mechanism is proposed, which allows one stream to extract information from another one. Extensive experiments have been conducted, and GeoMFormer shows significant improvements over baselines across all six tasks considered. Ablation studies clearly demonstrate the effectiveness of each module in the model.

**Strengths:**

1. This paper is well-written and well-motivated.
2. The proposed architecture has a clear and consistent design principle, resulting in an elegant and effective model that performs even better than those utilizing complex features.
3. The design of cross-attention is intuitive but effective in preserving more structural information.
4. Extensive experiments have clearly demonstrated the superiority of GeoMFormer over many competitive baselines.

**Weaknesses:**

1. I'm generally satisfied with the extensive experiments, but it would be better to benchmark GeoMFormer on QM9 due to its popularity.
2. For experiments on PCQM4MV2, it seems only GeoMFormer and Uni-Mol+ are directly comparable, as both of them essentially utilize more supervision information.

**Questions:**

No.

---

> ### Author Response · Authors · 2023-11-22
> **Response to Reviewer awZP**
>
> Thank you for your recognition of the effectiveness and significance of our work. We also appreciate your comments to further improve our work. Here are our responses to your questions.
>
> > **Regarding experiments on the QM9 dataset**
>
> Thank you for the suggestion. We also agree with you that the QM9 dataset has been adopted as a popular benchmark to evaluate the effectiveness of geometric molecular models. This dataset provides quantum chemical properties (e.g., HOMO-LUMO gap, total energy, etc) at Density Functional Theory level for a relevant, consistent, and comprehensive chemical space of 134k small organic molecules with equilibrium structures. Thus, the tasks of the QM9 dataset are invariant prediction tasks. In our experiments, we actually adopted several popular datasets that evaluate molecular model in a similar way. For example, the Molecule3D dataset consists of around 3.8 million molecules in total and requires the model to predict the HOMO-LUMO energy gap given the equilibrium structures, which is similar to the QM9 task and is much larger. Besides, the experiments on MD17 also examine the model's ability to perform the energy prediction task, which is also very popular. Moreover, we also follow your advice to further conduct experiments on the QM9 dataset. Due to the time and computational resources limits, we will update the results on the QM9 dataset to further improve the quality of our work.
>
> > **Regarding baselines on the PCQM4Mv2 dataset**
>
> We would like to clarify the setting of the PCQM4Mv2 dataset. In the training set of the PCQM4Mv2 dataset, each molecule is associated with both its 2D graph structure and 3D geometric structure obtained by using DFT-level relaxation for equilibrium states. In the validation set, each molecule is associated with only its 2D graph structure. It is a reasonable setting because the DFT-level relaxation is extremely expensive. In real-world applications, if a deep learning model need to take expensive equilibrium structures as the input, it has no more advantages compared to classical methods when we need to perform prediction on new molecules. Different from previous baselines focusing on graph learning, recent works including Transformer-M, Uni-Mol+ have noticed the value of these 3D equilibrium structures provided in the training time and pushed the frontier of this challenging task. In our experiment, we follow the training strategy of Uni-Mol+. Compared to these strong baselines, our framework is demonstrated to be effective by the empirical results and remains efficient compared to the sophisticated Uni-Mol+ architecture. From these points, we believe the PCQM4Mv2 task is a good evaluation protocal.
>
> We thank you again for your efforts in reviewing our paper. We hope the clarifications can address your questions and we will update new results following your suggestions.

---

> > ### Comment · Reviewer_awZP · 2023-11-23
> >
> > I have read the authors' responses and other reviews. Overall, I like this work and I will keep my rating towards acceptance.

---

> > > ### Author Response · Authors · 2023-11-23
> > > **Thank you for the timely feedback**
> > >
> > > We sincerely appreciate your timely feedback. We're pleased to hear that you appreciate our efforts and that you are inclined towards accepting our work. Your recognition of the significance of our work is encouraging. We will carefully revise our paper and update these new results following all reviewers' useful suggestions to improve the quality of our work. Again, we sincerely thank you for your thoughtful review and positive feedback.

---

### Official Review · Reviewer_qDQg · 2023-11-07

**Soundness:** 3 good
**Presentation:** 3 good
**Contribution:** 3 good
**Rating:** 6
**Confidence:** 3

**Summary:**

The authors propose a novel molecular model called GeoMFormer for molecular modeling, which is based on the Transformer architecture. GeoMFormer incorporates two distinct streams for preserving invariant and equivariant representations and employs cross-attention modules to facilitate information exchange between them. The proposed GeoMFormer is a general and adaptable framework, with other existing architectures being special cases. Extensive experiments demonstrate GeoMFormer's strong performance on invariant and equivariant tasks across various types and scales. The potential of our GeoMFormer can be further explored in a broad range of applications in molecular modeling.

**Strengths:**

GeoMFormer divides the process of learning invariant and equivariant representations into distinct streams, which are interconnected through self-attention and cross-attention modules. Through these cross-attention modules, the invariant stream is enriched with structural information from the equivariant stream, while the equivariant stream benefits from non-linear transformations originating from the invariant stream. This enables the comprehensive modeling of interatomic interactions, both within individual feature spaces and across them, in a unified and simultaneous fashion.


The proposed architecture is general. Many established methods can be seen as specific instances within our framework. For instance, PaiNN (Schütt et al., 2021) and TorchMD-NET (Thölke & De Fabritiis, 2022) can be configured as distinct realizations by adhering to the design principles of GeoMFormer and selecting appropriate configurations for essential building components.



The experimental results are thorough, including invariant and equivariant tasks, covering multiple mainstream tasks in the molecule modeling. The proposed GeoMFormer was compared with a bunch of state-of-the-art methods and outperform them in various setups.



The paper is well-written and easy to follow.

**Weaknesses:**

No.

**Questions:**

I look forward to more discussion of application of GeoMFormer in many other areas outside molecule modeling.



In the experiment, I wonder why different invariant tasks use different neural architectures as baseline methods, e.g., Section 5.2 and 5.3. The same thing happens for equivariant tasks.


I wonder if the author could report the results of multiple runs, e.g., standard deviation to enhance the empirical results.


In N-BODY SIMULATION in Section 5.4, what are the input feature and groundtruth? input is 3D position, output is velocity? Please add more details. Thanks!

---

> ### Author Response · Authors · 2023-11-22
> **Response to Reviewer qDQg (Part 1/2)**
>
> Thank you for your recognition of the effectiveness and significance of our work. Here are our responses to your questions:
>
> > **Provide more discussion on application of GeoMFormer outside molecular modeling**
>
> Thank you for the suggestion. Our primary focus on molecular representation learning stems from its pivotal role in various real-world applications with great significance, e.g., drug discovery. In fact, our work provides a unified framework to simultaneously perform invariant and equivariant tasks in an effective manner, which is not limited only to molecular modeling.
>
> In the scientific domain, our methodology can be naturally extended to other critical applications like protein biology, material design and so on. Although objects of interests in these applications may have different formulations from molecular systems used in our work, they generally exhibit symmetries and require models to perform invariant and equivariant tasks well. Take the OC20 as a demonstrating example. Different from small molecules, data points in OC20 are catalyst-adsorbate complexes. Hence, we need to incorporate the Periordic Boundary Conditions (PBC) into our model, while the invariant and equivariant constraints are naturally satisfied with our framwork. In this scenario, our framework consistently achieves strong performance across invariant and equivariant tasks. It is noteworthy that such data priors are also important in applications like material design, which indicates the potential of our framework in these challenging tasks.
>
> Moreover, our framework can also be applied to 3D vision, which also exhibits symmetries to the SE(3) group. Typical tasks including classification (invariant), part segmentation (invariant), neural implicit reconstruction (equivariant), visual robotic manipulation (equivariant) and so on. As we can see, these tasks also require the model to well perform invariant and equivariant prediction, and the format of 3D point clouds is also similar to molecules. Therefore, it is naturally to generalize our GeoMFormer to this significant domain.

---

> ### Author Response · Authors · 2023-11-22
> **Response to Reviewer qDQg (Part 2/2)**
>
> > **Regarding baseline methods in different tasks**
>
> To ensure a fair comparison, we follow previous works to choose strong baselines for each dataset and reported the official results from the original paper if available for most of the experiments. In fact, few previous works have benchmarked their proposed model across all the extensive and diverse datasets presented in our paper. Due to this, we have different baselines in different tasks like Section 5.2 and 5.3. This phenomenon further demonstrates the generality and effectiveness of our framework, as our GeoMFormer is able to simultaneously perform well on a wide range of tasks involving both invariant and equivariant prediction and consistently achieves strong performance compared to these specialized baselines.
>
> > **Provide results of multiple runs**
>
> Thank you for the suggestion. Tasks in our experiments can be divided into two categories according to their dataset sizes: (1) large-scale datasets, including OC20, PCQM4Mv2 and Molecule3D; (2) small-scale datasets, including N-Body simulation and MD17. For large-scale datasets, we observe that the results are rather stable across different runs, i.e., the standard deviation is extremely small. For small-scale datasets, we actually follow the convention of previous works to perform multiple runs for fair comparisons. For N-body simulation, we perform five runs with different random seeds and report the averaged results across these five runs. For MD17, we averaged the results over three random splits. Following the convention of previous works, we report the averaged results for both the datasets.
>
> > **Provide more details on the N-Body simulation experiment**
>
> Thank you for the suggestion. We provide more details here. In the N-Body simulation experiment, each data point is a particle system, which consists of 5 particles that carry a positive or negative charge and have an initial position and an initial velocity. The system is controlled by physical rules involving attractive and repulsive forces. The task requires the model to predict the positions of the five particles after 1,000 timesteps. Specifically, the input feature includes (1) the initial positions of the five particles $p^0=\{p^0_1,p^0_2,\cdots,p^0_5\}\in\mathbb{R}^{5\times 3}$ , (2) the initial velocities of five particles $v^0=\{v^0_1,v^0_2,\cdots,v^0_5\}\in\mathbb{R}^{5\times 3}$ and (3) the charges (positive or negative) of five particles $c=\{c_1,c_2,\cdots,c_5\}\in\{-1,1\}^5$. The output is the predicted positions of the five particles after 1,000 timesteps $\hat{p}^t=\{\hat{p}^t_1,\hat{p}^t_2,\cdots,\hat{p}^t_5\}\in\mathbb{R}^{5\times 3}$ and the groundtruth is the true positions of the five particles after 1,000 timesteps $p^t=\{p^t_1,p^t_2,\cdots,p^t_5\}\in\mathbb{R}^{5\times 3}$ . we will incorporate additional details in the revised version of the paper.
>
> We thank you again for your efforts in reviewing our paper. We have carefully replied to each of your questions, and we sincerely look forward to your re-evaluation of our submission based on our clarifications. May you have any further questions, please tell us and we are willing to address your concerns.

---

### Official Review · Reviewer_vMxo · 2023-11-09

**Soundness:** 3 good
**Presentation:** 3 good
**Contribution:** 1 poor
**Rating:** 5
**Confidence:** 4

**Summary:**

This paper proposes a new  Transformer-based molecular model, GeoMFormer based on standard Transformer modules. The architecture has an invariant representation brach and an equivariant representation brach. Cross-attentions are used to fuse the two kinds of representations in the architecture. Extensive experiments are conducted on many datasets. The model outperforms baselines.


------- After rebuttal -----

I would like to raise my score from "reject" to "marginally below the acceptance threshold" beacause of the new ablation experiment on MD17.

In my view, this paper still does not meet the standards required for ICLR because its motivation is not clear. The central argument of this paper is "effectively learning both invariant representations and equivariant representations simultaneously". As a study focused on information fusion, it is pivotal to analyze the distinct characteristics of these two types of information. Such an analysis is vital to demonstrate the importance of their fusion. Unfortunately, this paper presents a superficial exploration of these differences, which undermines its overall quanlity.

In addition, some experiments do not well support the core contribution of this paper. For instance, in the ablation studies in Table 8, the model without any cross-attn can still achieve competitive performance. The cross-attn module is the main technological contribution of this paper.

**Strengths:**

S1: The proposed model is validated on extensive experiments and tasks;

S2. The performance of the model is commendable;

S3: The paper is well written.

**Weaknesses:**

W1: The intuition and rationale of modeling is not clear. There is a lack of analysis of the difference between an invariant representation and an equivariant representation in terms of the information they contain.

W2: The novelty in this paper is limited. Simultaneously, using an invariant representation and an equivariant representation, such as VisNet, is not new. Cross-attention is well-known in multimodal deep learning.

W3: The experimental results cannot support the advantage of fusing two kinds of representation. The ablation studies in Table 8 are interesting. The model without any cross-attn can still achieve competitive performance. The cross-attn module, the main contribution of this paper, may contribute little to performance in experiments.

**Questions:**

Q1: Can you give empirical evidence that invariant and equivariant representations have different useful information for molecular learning?

Q2: In Table 8, the model without cross-attn can still achieve competitive performance. From this point, is it very important to fuse two kinds of representation on such a powerful Transfermor model?

---

> ### Author Response · Authors · 2023-11-22
> **Response to Reviewer vMxo (Part 1/5)**
>
> Thank you for spending time reviewing our paper. Here are our responses to your questions:
>
> > **Provide analysis of the difference between invariant and equivariant representations in terms of the information they contain (W1 & Q1)**
>
> In the Introduction section of our paper, we actually provide discussion of the strengths/weaknesses between modeling invariant and equivariant representations, which demonstrates the necessity of developing a powerful model that can perform both invariant and equivariant prediction with strong performance at the same time. Here we would like to further elucidate the disparity between invariant and equivariant representations in terms of the information they encapsulate, substantiated by additional supporting evidence.
>
> From the view of neural network design, equivariant representations preserve more complete structural information, but the equivariant constraints restrict the choice of operations that can be used. On the other hand, invariant representations offer more flexibility for non-linear operations, but retain less structural information.
>
> First, we provide more explantions and supporting evidence on why equivariant representations preserve more complete structural information. In the literature, existing invariant models commonly use invariant features of molecules (e.g., interatomic distances, bond angles, dihedral/torsion angles, improper angles and so on) which are extracted from the raw 3D geometric structures of molecules (the position of each atom in the molecule). Although these invariant features are important to describe the properties of molecules, the mapping from raw geometric structures to invariant features still introduces information loss:
> - In a recent study on the expressive power of geometric graph neural networks [1], the authors provided thorough theoretical analysis on the expressive power of invariant layers and equivariant layers commonly used in geometric graph neural networks by leverging the Geometric Weisfeiler-Lehman test. Each invariant layer in geometric graph neural networks corresponds to the proposed Invariant Geometric Weisfeiler-Lehman test (IGWL). In their framework, previous mentioned invariant features can be categorized into different classes in terms of the least number of nodes (atoms in the context of molecular modeling) that are involved to compute it. For example, interatomic distances involve two atoms, while bond angles involve three atoms. This features are then called as $k$-body scalars. Based on this framework, the authors carefully characterized the expressive power of invariant layers and prove that "any number of iterations of IGWL cannot distinguish any 1-hop identical geometric graphs $\mathcal{G}_1$ and $\mathcal{G}_2$ where the underlying attributed graphs are isomorphic" and "GWL (corresponds to equivariant layer) can distinguish by propagating information from the geometric structure", which indeed indicates the information loss from raw geometric structures to invariant features.
> - In PaiNN [2], the authors also provided intuition on the point that there exist limits of invariant representations. Firstly, the authors demonstrated that to express enough information (like bond angles of all neighbors of an atom), using invariant representations would requires computations with higher complexity than using equivariant representations (see the analysis of Table 1 in [2]). Moreover, the authors also showed that equivariant representations allow to propagate crucial geometrical information beyond the neighborhood, which is not possible in the invariant case (see the analysis of Figure 1 in [2]). These evidence motivated the authors to develop models based on equivariant representations.

---

> ### Author Response · Authors · 2023-11-22
> **Response to Reviewer vMxo (Part 2/5)**
>
> Second, we also demonstrate that invariant representations allow more flexibility for non-linear operations while the equivariant constraints restrict the choice of equivariant operations that can be used. For invariant representations, there exists no constraints on the operations that can be used. The invariant constraints are naturally satisfied once the invariant features are extracted. Hence, we can freely choose different operation designs with arbitrary non-linearity to increase the model capacity and incorporate priors for targeted tasks. However, it is not the same for equivariant representations. To satisfy the equivariant constraints, operation classes that are qualified to correctly process equivariant representations are restricted, and non-linear transformations commonly used in neural networks would break the constraints on equivariant representations. That is to say, although equivariant representations contain more complete information, it is restricted for models to well process and transform these information:
> - Most existing works use (1) vector operations; (2) tensor products of irreducible representations to develop equivariant operations. For the former one, the non-linearity is constrained. For the latter one, the computational complexity is high which impedes the deployment to large 3D systems.
> - Empirically, previous equivariant models cannot consistently achieve superior performance on invariant tasks compared to invariant models, due to the restricted non-linearity or computational complexity. For example, previous invariant models are still dominant in invariant prediction tasks on PCQM4Mv2 and Molecule3D (Table 4 and 5). Moreover, the model capacity of existing equivariant models is thus limited, and the ability to scale model size up is also restricted. In our preliminary experiments on previous models like PaiNN/TorchMD-Net, it is hard to train these models with larger model size, on which we observed the training instability issues.
>
> Based on the above explanations and evidence, we can see that invariant and equivariant representations are indeed different in terms of the information they contain. Compared to invariant representations, equivariant representations contain *more complete information* on the geometrical structures. Compared to equivariant representations, the information contained by invariant representations can be *better transformed and processed* with more flexible design choices of operations. Therefore, it is important to effectively learn both invariant representations and equivariant representations at the same time, towards which our work provides a powerful and unified framework.
>
> [1] Joshi, C. K., Bodnar, C., Mathis, S. V., Cohen, T., & Lio, P. (2023). On the expressive power of geometric graph neural networks. ICML 2023.
>
> [2] Schütt, K., Unke, O., & Gastegger, M. (2021, July). Equivariant message passing for the prediction of tensorial properties and molecular spectra. In International Conference on Machine Learning (pp. 9377-9388). PMLR.

---

> ### Author Response · Authors · 2023-11-22
> **Response to Reviewer vMxo (Part 3/5)**
>
> > **Regarding the novelty of the paper (W2)**
>
> We would like to clarify that the novelty of our work goes beyond merely using invariant and equivariant representations and cross-attention modules. The key innovation of our work lies in the proposed general design philosophy of our framework: *effectively learning both invariant representations and equivariant representations simultaneously, and completely modeling interatomic interactions within/across feature spaces separately [invariant→invariant interactions, equivariant→equivariant interactions, invariant→equivariant interactions, equivariant→invariant interactions]*. This holistic modeling is unified within a general and scalable framework by using the attention mechanism (self-attn and cross-attn) with constraints in the two-stream Transformer.
>
> From this perspective, our work stands out in its novelty, as existing models often fall short in crucial aspects:
> - Do not maintain or iteratively update both invariant and equivariant representations throughout the models;
> - Do not completely model all the interactions (lacking interatomic or self/cross feature space propagation), or entangle different interactions instead of separation;
> - Instantiate these interactions by using operations case-by-case instead of unification.
>
> The importance of complete modeling interactions and unified modeling is *firstly highlighted by our work* and its effectiveness in molecular modeling *is well supported by the provided significant results* (as you also appreciate the achieved performance of our GeoMFormer model). From these clarifications, the novelty of our contributions should be taken into account.
>
> We also like to thank the reviewer for pointing out the VisNet model [1]. Here we provide more discussion. In the VisNet framework, the model also maintains both invariant and equivariant representations, and uses the proposed Runtime Geometry Calculation (RGC) that can efficiently incorporate information from angles. VisNet also demonstrates good performance on the quantum property prediction task like QM9.
>
> However, our GeoMFormer is different from VisNet. From the perspective of our framework, the VisNet model do not have pure and interatomic equivariant→equivariant interactions and invariant→invariant interactions. The entanglement of all kinds of interactions makes the messages from different interactions contribute unbalancedly to invariant and equivariant features. Besides, most experiments of VisNet are invariant prediction tasks. In the MD17 experiment, the force prediction is performed by using the negative gradients of the predicted total potential energy with respect to the atomic coordinates. In real-world applications, there are indeed other significant equivariant prediction tasks like structure prediction (e.g., OC20 IS2RS) and dymanic system simulation (N-Body simulation) that require the model to directly output equivariant predictions. In our work, we comprehensively verify the ability of our GeoMFormer to perform different kinds of invariant and equivariant tasks, which further demonstrates the significance and effectiveness of our framework. Overall, we think the VisNet paper is a good reference to make our related works more complete, and we will definitely cite the VisNet paper and carefully revised our paper to add more discussion on it.
>
> [1] Wang, Y., Li, S., He, X., Li, M., Wang, Z., Zheng, N., ... & Wang, T. (2022). ViSNet: an equivariant geometry-enhanced graph neural network with vector-scalar interactive message passing for molecules (No. arXiv: 2210.16518).

---

> ### Author Response · Authors · 2023-11-22
> **Response to Reviewer vMxo (Part 4/5)**
>
> > **Regarding the ablation study (W3 & Q2)**
>
> We would like to clarify the significance of our ablation study (Table 8) with updated experiment results. As can be seen from the baseline performance of the N-Body simulation task, the absolute number of the MSE criterion is very low, which means that we should focus on the relative error reduction when comparing the performance of two models. In Table 8, we actually have the following conclusions on the ablation study: (1) removing any cross-attn module (Inv-Cross-Attn, Equ-Cross-Attn) consistently induces performance drop; (2) the gains brought by cross-attn modules are significant: 7.8% relative improvement for Inv-Cross-Attn, 13.0% relative improvement for Equ-Cross-Attn, 17.5% relative improvement for both Inv-Cross-Attn and Equ-Cross-Attn.
>
> Moreover, we further conducted ablation study on the MD17 benchmark as a double-check. The results are presented in the following table. From these results, we can see the same trend as in the N-Body simulation task: (1) removing any cross-attn module (Inv-Cross-Attn, Equ-Cross-Attn) consistently induces performance drop; (2) the gains brought by cross-attn modules are extremely significant for energy prediction (invariant): 18.7% relative improvement for Inv-Cross-Attn, 9.8% relative improvement for Equ-Cross-Attn, 20.8% relative improvement for both Inv-Cross-Attn and Equ-Cross-Attn; (3) the gains brought by cross-attn modules are also extremely significant for force prediction (equivariant): 28.0% relative improvement for Inv-Cross-Attn, 43.9% relative improvement for Equ-Cross-Attn, 60.8% relative improvement for both Inv-Cross-Attn and Equ-Cross-Attn.
>
> Based on these thorough results, we indeed demonstrate the significance and effectiveness of our framework, which is strongly supported and should be definitely taken into account.

---

> ### Author Response · Authors · 2023-11-22
> **Response to Reviewer vMxo (Part 5/5)**
>
> **Table 1. Ablation Studies on MD17.**
>
> **Energy**:
>
> | Inv-Self-Attn | Inv-Cross-Attn | Equ-Self-Attn | Equ-Cross-Attn |  Aspirin  |  Benzene  |  Ethanol  | Malondialdehyde | Naphthalene | Salicylic Acid |  Toluene  |  Uracil   |
> | :-----------: | :------------: | :-----------: | :------------: | :-------: | :-------: | :-------: | :-------------: | :---------: | :------------: | :-------: | :-------: |
> | $\checkmark$  |  $\checkmark$  | $\checkmark$  |  $\checkmark$  | **0.118** | **0.052** | **0.047** |    **0.071**    |  **0.081**  |   **0.099**    | **0.078** | **0.095** |
> |   $\times$    |  $\checkmark$  | $\checkmark$  |  $\checkmark$  |   0.156   |   0.063   |   0.058   |      0.085      |    0.092    |     0.115      |   0.090   |   0.102   |
> | $\checkmark$  |    $\times$    | $\checkmark$  |  $\checkmark$  |   0.161   |   0.062   |   0.060   |      0.086      |    0.111    |     0.113      |   0.094   |   0.101   |
> | $\checkmark$  |  $\checkmark$  |   $\times$    |  $\checkmark$  |   0.131   |   0.059   |   0.052   |      0.081      |    0.084    |     0.104      |   0.085   |   0.097   |
> | $\checkmark$  |  $\checkmark$  | $\checkmark$  |    $\times$    |   0.143   |   0.056   |   0.051   |      0.078      |    0.097    |     0.106      |   0.081   |   0.099   |
> | $\checkmark$  |    $\times$    |   $\times$    |  $\checkmark$  |   0.169   |   0.062   |   0.061   |      0.087      |    0.113    |     0.115      |   0.094   |   0.103   |
> | $\checkmark$  |    $\times$    | $\checkmark$  |    $\times$    |   0.172   |   0.064   |   0.061   |      0.086      |    0.112    |     0.116      |   0.095   |   0.103   |
> |   $\times$    |  $\checkmark$  |   $\times$    |  $\checkmark$  |   0.184   |   0.069   |   0.064   |      0.094      |    0.121    |     0.124      |   0.102   |   0.107   |
> |   $\times$    |  $\checkmark$  | $\checkmark$  |    $\times$    |   0.181   |   0.071   |   0.067   |      0.093      |    0.118    |     0.121      |   0.101   |   0.109   |
> | $\checkmark$  |    $\times$    |   $\times$    |    $\times$    |   0.193   |   0.076   |   0.071   |      0.097      |    0.126    |     0.129      |   0.105   |   0.113   |
>
>
>
> **Forces**:
>
> | Inv-Self-Attn | Inv-Cross-Attn | Equ-Self-Attn | Equ-Cross-Attn |  Aspirin  |  Benzene  |  Ethanol  | Malondialdehyde | Naphthalene | Salicylic Acid |  Toluene  |  Uracil   |
> | ------------- | -------------- | ------------- | :------------: | :-------: | :-------: | :-------: | :-------------: | :---------: | :------------: | :-------: | :-------: |
> | $\checkmark$  | $\checkmark$   | $\checkmark$  |  $\checkmark$  | **0.171** | **0.146** | **0.062** |    **0.133**    |  **0.040**  |   **0.098**    | **0.041** | **0.068** |
> | $\times$      | $\checkmark$   | $\checkmark$  |  $\checkmark$  |   0.223   |   0.152   |   0.086   |      0.162      |    0.051    |     0.117      |   0.062   |   0.079   |
> | $\checkmark$  | $\times$       | $\checkmark$  |  $\checkmark$  |   0.257   |   0.159   |   0.104   |      0.178      |    0.063    |     0.126      |   0.076   |   0.091   |
> | $\checkmark$  | $\checkmark$   | $\times$      |  $\checkmark$  |   0.292   |   0.167   |   0.143   |      0.311      |    0.079    |     0.203      |   0.096   |   0.134   |
> | $\checkmark$  | $\checkmark$   | $\checkmark$  |    $\times$    |   0.281   |   0.160   |   0.167   |      0.272      |    0.094    |     0.185      |   0.081   |   0.113   |
> | $\times$      | $\checkmark$   | $\checkmark$  |    $\times$    |   0.313   |   0.164   |   0.196   |      0.319      |    0.134    |     0.191      |   0.105   |   0.195   |
> | $\checkmark$  | $\times$       | $\checkmark$  |    $\times$    |   0.366   |   0.187   |   0.212   |      0.352      |    0.159    |     0.264      |   0.161   |   0.237   |
> | $\times$      | $\checkmark$   | $\times$      |  $\checkmark$  |   0.324   |   0.173   |   0.189   |      0.331      |    0.129    |     0.237      |   0.127   |   0.167   |
> | $\checkmark$  | $\times$       | $\times$      |  $\checkmark$  |   0.358   |   0.182   |   0.219   |      0.337      |    0.172    |     0.272      |   0.134   |   0.241   |
> | $\times$      | $\times$       | $\checkmark$  |    $\times$    |   0.411   |   0.194   |   0.227   |      0.361      |    0.188    |     0.289      |   0.173   |   0.258   |
>
> Thank you again for reviewing our paper. We have carefully replied to each of your questions and provided updated results. We look forward to your re-evaluation of our submission based on our responses and updated results.

---

> ### Comment · Reviewer_vMxo · 2023-11-23
> **Thanks for your reply**
>
> Thanks for your reply.
>
> (1) Your statement regarding the difference between invariant representations and equivariant representations --- "Although these invariant features are important to describe the properties of molecules, the mapping from raw geometric structures to invariant features still introduces information loss" --- may not be correct. In chemistry, the internal coordinate system, which only use invariant features like distances and angles, is widely used. This system effectively and completely describes 3D molecular structures, without any information loss. The case presented in the "expressive power of geometric graph neural networks" can be addressed using long-range invariant feature or incorporating a non-coplanar virtual node.
>
> (2) The central argument of this paper is  "effectively learning both invariant representations and equivariant representations simultaneously". As a study focused on information fusion, it is pivotal to analyze the distinct characteristics of these two types of information, so as to show the significance of fusion. Without this differentiation, the paper may not be convincing and lacks a well-defined motivation.
>
> (3) The new additional ablation studies on MD17  are commendable.

---

### Meta-Review · Area_Chair_7ZVm · 2023-12-09

**Metareview:**

This paper studies molecular representation learning by combining invariant and equivariant representations. The proposed methods achieves promising performance, but a major concern is that the paper does not clearly demonstrate that the improved performance is the result of fusing invariant and equivariant representations. In addition, other parts of experiments are not consistent with the proposed claims. Thus a reject is recommended.

**Justification For Why Not Higher Score:**

The proposed methods achieves promising performance, but a major concern is that the paper does not clearly demonstrate that the improved performance is the result of fusing invariant and equivariant representations.

**Justification For Why Not Lower Score:**

NA

---

### Decision · Program_Chairs · 2024-01-16

Reject